# Research Progress and Development of Near-Infrared Phosphors

**DOI:** 10.3390/ma16083145

**Published:** 2023-04-16

**Authors:** Tongyu Gao, Yuanhong Liu, Ronghui Liu, Weidong Zhuang

**Affiliations:** 1GRIREM Advanced Materials Co., Ltd., Beijing 100088, China; 2National Engineering Research Center for Rare Earth Industry, GRINM Group Co., Ltd., Beijing 100088, China; 3General Research Institute for Nonferrous Metals, Beijing 100088, China; 4School of Metallurgical and Ecological Engineering, University of Science and Technology Beijing, Beijing 100083, China; 5Beijing Key Laboratory for Green Recovery and Extraction of Rare and Precious Metals, University of Science and Technology Beijing, Beijing 100083, China

**Keywords:** NIR phosphor-converted LED (pc-LED), near-infrared (NIR) phosphor, luminescence property

## Abstract

Near-infrared (NIR) light has attracted considerable attention in diverse applications, such as food testing, security monitoring, and modern agriculture. Herein, the advanced applications of NIR light, as well as various devices to realize NIR light, have been described. Among the diverse NIR light source devices, the NIR phosphor-converted light-emitting diode (pc-LED), serving as a new-generation NIR light source, has obtained attention due to its wavelength-tunable behavior and low-cost. As one of the key materials of the NIR pc-LED, a series of NIR phosphors have been summarized depending on the type of luminescence center. Meanwhile, the characteristic transitions and luminescence properties of the above phosphors are illustrated in detail. In addition, the status quo of NIR pc-LEDs, as well as the potential problems and future developments of NIR phosphors and applications have also been discussed.

## 1. Introduction

Light is everywhere and closely related to everyone’s life. At present, light in the range of 400–700 nm is mainly used in general lighting and backlight display, while light with wavelengths beyond 700 nm is nonvisible light. According to the definition of the American Society for Testing and Materials Testing (ASTM), the near-infrared (NIR) wavelength range is between 780 and 2526 nm. Generally, the NIR region can be divided into two regions: short-wavelength NIR (780–1100 nm) and long-wavelength NIR (1100–2526 nm) [1,2].

As shown in Figure 1, the complementary light source in the field of modern agriculture and security monitoring, as well as the information detection light source in the field of food safety testing, require efficient NIR light [3,4,5]. For instance, spectra with a peak position at 850 nm have gained recent attention for great applications in security monitoring, face recognition, automobile sensors, etc., while the light source, with a spectrum peaked at 940 nm, is expected to be used in blood oxygen detection, as well as touch screens. 

## 2. Analysis of Application Fields

### 2.1. Modern Agriculture

Regardless of weather, season, or time of day, plants much like humans, need light to thrive. Light plays a role in plant growth mainly in two aspects: it participates in photosynthesis as energy in order to promote the accumulation of plant energy, and it serves as a signal to stimulate phytonutrient accumulation and regulate the growth and development of a plant, such as in germination, flowering, and stem growth [6]. Infrared light, especially NIR light, has a great influence on chlorophyll A, B, and phytochrome Pr and Pfr (exhibiting absorption at 420–500 nm violet–blue light and 660–780 nm far-red–NIR light) [7]. At present, LED light sources are widely used in plant factories for interplant irradiation to promote plant growth.

In particular, far-red light, with a peak position of 730 nm, has a “shading effect” on plants. Under the irradiation of this light, plants will feel that they are blocked from direct light and will work harder to grow to break through the “shading effect” [8]. Consequently, affecting the phytochrome photoreceptors in plants will encourage plants to grow taller and stronger.

### 2.2. Security Monitoring

At present, security monitoring products require not only improvements in the presentation of video images but also in all-weather observation requirements, with multidimensional, all-weather, three-dimensional, and intelligent development trends, to improve social management capabilities. With the continuous development of social security demands, the demand for security monitoring has been growing steadily, especially in the high-power infrared market.

Security monitoring requires NIR light sources with different wavebands of 700–940 nm for different monitoring scenarios. Currently, infrared light sources of 850 nm and 940 nm are widely adopted by night-vision cameras, and the emission spectra of such infrared light sources should be broad for effective performance [9]. Among them, the infrared light source centered at 850 nm has become mainstream in night-vision technology owing to its superior sensitivity to complementary metal-oxide semiconductor (CMOS) sensors, high radiometric power output, and long radiometric distance [10].

### 2.3. Food/Medical Testing

With the vigorous development of medicine, biochemistry, and other industries, people pay more and more attention to health care. Recently, health testing instruments and food composition testing instruments have gradually developed toward smaller sizing and portability [3]. Wearable food/medical testing devices loaded with NIR LEDs can be applied to sports training, home care, driving testing, and other fields. Because of the wide absorption and reflection spectrum range of organic elements in food and human organs, an NIR light source with a wide emission spectrum is generally needed in emerging applications such as food testing and health monitoring [4]. Herein, the most important aspect of light sources for NIR spectroscopy is to cover as broad a wavelength range of emitted light as possible. The larger the range of NIR light, the greater the number of objects that can be analyzed.

To determine ingredients or content, the target object is illuminated with infrared light of a wide wavelength range (usually 650 to 1050 nanometers). Parts of this light are reflected, and others are absorbed. This ratio varies from object to object, resulting in a unique molecular fingerprint for each item. The reflected light is collected by a special detector. Then, the software processes this data, compares it with documented information stored in the cloud, and finally, produces the measurement results [11,12]. For instance, it is found that NIR light sources, with peak emissions at 1180 nm and 1400 nm, can be effectively absorbed by glucose, which can be used for noninvasive blood glucose detection by using the above principle [13].

## 3. Introduction of Light-Emitting Device

### 3.1. Incandescent Bulb

The incandescent bulb, with a history of nearly 130 years, is the earliest mature artificial electric light source, which uses a filament heated to an incandescent state to emit light through the principle of thermal radiation. An incandescent bulb can provide continuous light emission from visible light to the NIR range. However, it suffers from short life, high energy consumption, low luminous efficiency, and high operating temperature, which limits its application. At the same time, as countries promote energy conservation, emissions reductions, environmental protection, and other policies, incandescent bulbs gradually fade out of the stage of history [14].

### 3.2. Halogen Lamp

The halogen lamp is a variation of an incandescent lamp, which can be produced by adding a halogen gas, such as bromine or iodine, to the bulb, which reacts with a tungsten filament sublimated at high temperatures. Similar to incandescent bulbs, halogen lamps provide a continuous light emission in the visible to NIR range and emit light more efficiently than incandescent bulbs [15]. However, when a halogen lamp needs to filter light in the process of presenting NIR light, most of the light is split, consequently resulting in low efficiency. In addition, halogen lamps generate a lot of heat in the process of use, as well as their large size and short service life, which limits their application.

### 3.3. (Al, Ga) As-Based LED

The (Al, Ga) As-based LED is an integrated light-emitting device that converts electrical energy into NIR light directly. NIR LED is made by a p-n junction such as (Al, Ga) As. It exhibits light by passing current into the p-n junction, which leads to the composite reaction of holes. The p-n junction then releases photons to obtain NIR light. So far, the AlGaAs-based LEDs have been the mainstream commercial device to achieve light ranging from far-red to NIR wavelengths [16]. However, the electroluminescence principle of the p-n junction determines that it can only emit monochromatic light with a narrow emission band (FWHM ± 40 nm), and cannot observe a continuous emission spectrum. To obtain broadband luminescence, multiple chips need to be combined, causing a complicated encapsulation process. Moreover, the AlGaAs material has demonstrated a severe thermal quenching behavior due to the thermal activation of electron-hole pairs. In addition, the cost of NIR LED chips is more than 10 times that of blue LED chips, which limits their application and popularization [17,18].

### 3.4. NIR Phosphor-Converted LED (NIR pc-LED)

The NIR phosphor-converted LED (NIR pc-LED), as an ideal and new generation of broadband output far-red/NIR-emitting light source [19,20], has gained recent attention for great applications [21,22] based on low-cost, superior efficiency, a wavelength-tunable behavior, and the mature commercialization of UV/blue chip [23,24]. NIR pc-LEDs have been developed by OSRAM [25], whose spectral composition can be easily controlled by employing blue LED chips with different kinds of broadband NIR phosphors, which has a wide application prospect in mobile phones and NIR spectrometers, but has not been applied in practice. Accordingly, a high-performance NIR phosphor, converting the UV/blue LED chip emissions to NIR luminescence is necessary for the NIR pc-LED, which needs to be developed urgently.

Related material systems, preparation technologies, and applications have been explored at home and abroad, mainly focusing on theoretical research, and progress has been serially made. Foreign companies such as OSRAM and Mitsubishi Chemical have carried out technology and patent layouts for LED light source technologies and supporting luminous materials. A series of research institutions in China, such as GRIREM Advanced Materials Co., Ltd. (Beijing, China), Xiamen University, and other institutions have carried out the basic research and design of NIR luminescence materials. However, the progress in application exploration is slow and needs to be optimized.

## 4. Research Status of NIR Phosphor

In recent years, a wealth of NIR phosphors have been extensively studied. The emission spectral position of NIR phosphors can depend on the luminescence center. It is highlighted that the emission of a part of the NIR phosphors depends on the spin-forbidden 4f–4f transitions of lanthanide dopants, which exhibit narrow peaks [26], such as Pr^3+^(1037 nm: ^1^D_2_→^3^F_4_), Nd^3+^(886 nm: ^4^F_3/2_→^4^I_9/2_, 1064 nm: ^4^F_3/2_→^4^I_11/2_, and 1340 nm: ^4^F_3/2_→^4^I_13/2_), Ho^3+^(966 nm: ^5^F_5_→^5^I_7_, 1012 nm: ^5^F_4_→^5^I_6_, and 1180 nm: ^5^I_6_→^5^I_8_), Er^3+^(800 nm: ^4^I_9/2_→^4^I_15/2_, 980 nm: ^4^I_11/2_→^4^I_15/2_, and 1540 nm: ^4^I_13/2_→^4^I_15/2_), Tm^3+^(800 nm: ^3^H_4_→^3^H_6_ and 1800 nm: ^3^F_4_→^3^H_6_), and Yb^3+^(980 nm: ^2^F_5/2_→^2^F_7/2_). In addition, transition metal ions (Cr^3+^, Fe^3+^, and Mn^6+^) and main-group elements (such as Bi) can also achieve tunable NIR luminescence [27].

### 4.1. NIR Phosphor Activated by Lanthanides

The trivalent rare earth ion (RE^3+^) is one of the most studied luminescence centers for NIR luminescence, where the emission wavelength depends on the spin-forbidden 4f–4f transitions of lanthanide dopants. Consequently, excitation and emission of RE^3+^-doped phosphors are almost not affected by matrix structure and crystal field, that is, the peak positions are fixed. Therefore, the RE^3+^-doped NIR phosphors suffer from poor luminescence tuning performance, with sharp peaks. In addition, Eu^2+^ rare earth ion is considered a novel NIR luminescence center, with tunable luminescence behavior.

#### 4.1.1. NIR Phosphor Activated by Yb^3+^

As a rare earth element, Yb^3+^ has a relatively simple energy-level structure with only two energy levels, namely the ground state ^2^F_7/2_ and the excited state ^2^F_5/2_. Therefore, Yb^3+^ -activated phosphor cannot exhibit luminescence properties such as excited-state absorption or cross-relaxation [28].

Consequently, luminescence materials activated by Yb^3+^ are often applied in scintillators, pulsed solid-state lasers, medical NIR radiation therapy, and solid-state structure probes [29]. In addition, the f–f forbidden-transition of Yb^3+^ is less affected by the crystal field of the matrix, resulting in difficulty in emission spectrum adjustment. Since the band gap between the ground state ^2^F_7/2_ and the excited state ^2^F_5/2_ is small, the electrons located at the ground state can be directly excited to the excited state under a small energy excitation [30]. Therefore, researchers often use the energy transfer from a sensitizer to Yb^3+^ to excite the Yb^3+^-activated phosphors with ultraviolet light to obtain emission in the NIR region. When Yb^3+^ is doped with phosphor, it is easy to observe the typical emission of Yb^3+^ centered at 980 nm [31]. For instance, Yb^3+^ doped with molybdate Y_4_CdMo_3_O_16_ can achieve NIR luminescence in the range 900 to 1100 nm, with an absolute quantum efficiency of 15.35% (Figure 2a), which is ascribed to the ^2^F_5/2_→^2^F_7/2_ transition of Yb^3+^ ions [32]. Herein, the molybdate matrix Y_4_CdMo_3_O_16_ is cubic fluorite with space group *Pn-3n* (222). As shown in Figure 2b, the matrix first efficiently absorbs the near UV light (250–450 nm) and then transfers the energy to the Yb^3+^ ions (broadband downconversion from the matrix to the Yb^3+^ ions).

#### 4.1.2. NIR Phosphor Activated by Er^3+^

The NIR luminescence bands of Er^3+^-activated phosphors are 800 nm (^4^I_9/2_→^4^I_15/2_), 980 nm (^4^I_11/2_→^4^I_15/2_), and 1540 nm (^4^I_13/2_→^4^I_15/2_). The typical NIR luminescence at around 1540 nm contributes to the ^4^I_13/2_→^4^I_15/2_ transition, as well as the green emission owing to the ^4^S_3/2_→^4^I_15/2_ transition [33]. According to the energy-level diagram of Er^3+^, Er^3+^-activated phosphor can be excited by NIR photons (1510–1565 nm) with lower energy, and the phosphor exhibits upconversion luminescence at 545 nm (^4^S_3/2_→^4^I_15/2_), 665 nm (^4^I_9/2_→^4^I_15/2_), 800 nm (^4^I_9/2_→^4^I_15/2_), and 980 nm (^4^I_11/2_→^4^I_15/2_) [34]. Er^3+^-activated phosphors have played a key role in applications such as eye-safe lasers in fiber-optic communications, optical amplifiers in fiber-optic communications, and upconversion fluorescence with long afterglow [35]. On the other hand, Er^3+^ is also considered one of the most promising lanthanide ions to achieve NIR luminescence, and its emission wavelength matches the NIR-III biological window as well as the response curve of the InGaAs detector [36].

Setsuhisa Tanabe [37] et. al have reported a garnet persistent phosphor of Y_3_Al_2_Ga_3_O_12_ co-doped with Er^3+^ and Cr^3+^, which presents intense Cr^3+^ persistent luminescence (~690 nm) in the deep red region matching well with the first biological window (NIR-I, 650–950 nm), as well as Er^3+^ persistent luminescence (~1532 nm) in the NIR region matching well with the third biological window (NIR-III, 1500–1800 nm). Herein, the energy transfer from Cr^3+^ to Er^3+^ enhances the luminescence ranging from 1450 nm to 1670 nm of the Er^3+ 4^I_13/2_→^4^I_15/2_ transition (Figure 3). In addition, the afterglow can be lengthened by adding traps within the Cr^3+^ doping. After the blue light (460 nm) irradiation by 10 min, the long-duration persistent luminescence is more than 10 h, and its quantum efficiency is two times higher than that of the widely used ZnGa_2_O_4_:Cr^3+^ deep red persistent phosphor. This material can be excited by the commercial InGaAs blue light chip to obtain high-intensity and long-duration persistent luminescence. Additionally, the persistent luminescence nanoparticles are more conducive to biological probes, with improved optical resolution quality and deep tissue penetration performance, making it suitable for infrared third biological imaging windows.

Furthermore, the Er^3+^–Yb^3+^ co-doped phosphors have been widely studied for various applications. For instance, Er^3+^–Yb^3+^ co-doped phosphors are generally studied as upconversion luminescence materials. NaCeF_4_:Er^3+^/Yb^3+^ has been reported to present 1530 nm NIR emission under 980 nm excitation. Herein, according to Figure 4a, Yb^3+^ is considered the sensitizer to harvest 980 nm photons. Then, Er^3+^ can be elicited to the excited states via two or more successive energy transfers from Yb^3+^; consequently, the luminescence material can result in UC emission by radiative relaxation [38]. Enhanced green and red upconversion emissions are observed in Er/Yb-co-doped LiNbO_3_ under both 1550 nm and 980 nm excitation [39]. According to Figure 4b, three 1550 nm photons will excite the Er^3+^ from the ground ^4^I_15/2_ state to the green emitting ^2^H_11/2_ state via ground state absorption, excited-state absorption, and ESA3. The Er^3+^ exhibits a nonradiative relaxation from the ^4^I_9/2_ state to the ^4^I_11/2_ state, as well as presenting the red emission, owing to the ESA2 process. In addition, the two-photon process for populating the red-emittance may be attributed to the energy transfer. Therefore, the energy transfer and the energy back transfer processes are also responsible for the population of the red-emittance under 1550 nm excitation. The above work may provide greater probabilities to efficiently increase the solar spectrum response and further increase the photovoltaic efficiency of solar cells. La_3_Ga_5_GeO_14_: Cr^3+^/Yb^3+^/Er^3+^ phosphor can emit NIR luminescence at 976 nm and 1540 nm under the excitation of 660 nm. Herein, the 976 nm and 1540 nm NIR luminescence can be ascribed to Yb^3+^ and Er^3+^. When the concentration of Cr^3+^ is increased, the luminescence of Er^3+^ at 510–560 nm is inhibited, and Yb^3+^ acts as a “bridge” in the sample, which transfers energy from Cr^3+^ to Er^3+^ [40].

#### 4.1.3. NIR Phosphor Activated by Nd^3+^

According to the characteristic energy levels, NIR phosphor, with the lanthanide rare earth ion Nd^3+^, has absorption covering the UV to NIR region. Finally, Nd^3+^-doped phosphors present mainly three NIR luminescence ranges including 886–950 nm, 1050–1120 nm, and 1340 nm, which can be attributed to ^4^F_3/2_→^4^I_9/2_, ^4^F_3/2_→^4^I_11/2_, and ^4^F_3/2_→^4^I_13/2_ transitions, respectively. Accordingly, Nd^3+^-doped phosphors have important applications in various fields. For instance, the NIR luminescence centered at 1060 nm of Nd^3+^-doped phosphors can be commercially used in high-power solid-state lasers, and the NIR luminescence at 1340 nm has potential applications in optical amplifiers [41]. Lanthanide rare earth ion Nd^3+^ ions can be doped in the yttrium aluminum garnet Y_3_Al_5_O_12_ matrix (YAG) to obtain effective NIR light. The strengthened crystal field in LuAG: Nd^3+^ results in intense NIR luminescence centered at 808 nm, which can be applied in high-power solid-state lasers [42].

However, the luminescence of Nd^3+^ originates from transitions within the partially filled 4f levels, which is in principle spin-forbidden, resulting in weak and sharp excitations which are normally inefficient. Using the sensitizer, with allowed transitions, is an efficient way to solve the above problem. When Nd^3+^ is doped with Ca_2_PO_4_Cl, the phosphor exhibits NIR luminescence ranging from 800 to 1400 nm, which can be assigned to the characteristic transitions of Nd^3+^ ions. Further, the absorption peaks in the visible-NIR region extending from 450 to 850 nm are attributed to the f–f transitions of Nd^3+^. As shown in Figure 5 [43], it is interesting to note the appearance of the Eu^2+^ emission band in the excitation spectrum of Nd^3+^, which indicates the occurrence of an efficient energy transfer from Eu^2+^ to Nd^3+^. As a result, the improved luminescence properties of Ca_2_PO_4_Cl: Eu^2+^, Nd^3+^ are useful for Nd^3+^-based optical and photonic devices. 

While in Sr_2_Si_5_N_8_, the Nd^3+^ emission is sensitized to 450 nm light by co-doping with Eu^2+^, providing a large absorption cross section in the blue spectral region. The same transition can be observed in the PLE spectra of Nd^3+^ of the Eu^2+^ co-doped material, where it is responsible for the strongly red-shifted onset of the spectrum of Eu^2+^ at 600 nm, indicating the energy transfer from Eu^2+^ to Nd^3^+ [44]. Furthermore, Ce^3+^ is also an ideal sensitizer to enhance the luminescence of Nd^3+^ in various hosts. In the yttrium aluminum garnet (YAG) matrix, the PLE spectrum of Nd^3+^ matches well with the PL spectra of Ce^3+^ (Figure 5c) [45]. It is deduced that the energy transfer from Ce^3+^ to Nd^3+^ takes place. As shown in Figure 5d, after excitation of the 5*d *state of Ce^3+^, the Ce^3+^ electrons will relax to the 4*f *ground state, resulting in a broadband emission centered at 550 nm. Meanwhile, most electrons transfer to the ^2^G_7/2_ level of Nd^3+^. After relaxing to the ^4^F_3/2_ level through nonradiative decay, Nd^3+^ exhibits enhanced luminescence at 888 nm and 1064 nm.

#### 4.1.4. NIR Phosphor Activated by Ho^3+^

Thanks to the abundant 4f energy-level structure, Ho^3+^-doped luminescence material, with 966 nm (^5^F_5_→^5^I_7_), 1012 nm (^5^F_4_→^5^I_6_), and 1180 nm (^5^I_6_→^5^I_8_) NIR luminescence, is conducive to expanding the transmission capacity of optical networks and improving the optical response of solar cells with narrow bandgaps [46].

As shown in Figure 6, β-NaYF_4_:Ho^3+^ has been reported as an efficient quantum-splitting (QS) phosphor with three-step sequential three-photon near-infrared (NIR) quantum-splitting. The phosphor absorbs an ultraviolet photon, consequently splitting it into three NIR photons with wavelengths at 850, 1015, and 1180 nm. Herein, the ^3^K_8_, ^5^F_2_, and ^5^I_6_ electronic states of Ho^3+^ ions can be considered intermediate levels; the quantum-splitting (QS) phosphor exhibits a total QY of about 124%. The above work offers an interesting route toward the design of efficient photonic devices [47].

However, Ho^3+^-activated phosphors present low luminescence intensity owing to the complex energy-level pattern and forbidden-transition of Ho^3+^. Generally, Yb^3+^ is used as a sensitizer to achieve an intense NIR luminescence of Ho^3+^ near 1190 nm under 980 nm laser diode excitation [48]. Therefore, Ho^3+^ and Yb^3+^ are combined to form an energy clustering in the KLu_2_F_7_. When the phosphor is excited by UV light, it shows peaked NIR luminescence at 1190 nm. Meanwhile, the luminescence intensity can be enhanced by five times in the transfer from Yb^3+^ to Ho^3+^ [49].

#### 4.1.5. NIR Phosphor Activated by Tm^3+^

Tm^3+^-doped phosphors exhibit NIR luminescence peaked at 786 nm, 800 nm (^3^H_4_→^3^H_6_), and 1800 nm (^3^F_4_→^3^H_6_). When Tm^3+^ is doped in YNbO_4_ [50], it is found to possess intense two-photon, strong three-photon, and moderate four-photon quantum-cutting luminescence peaking at 1820 nm owing to the simultaneous transition of the Tm^3+^. According to Figure 7a, the up-limit of the two-, three-, and four-photon near-infrared quantum-cutting efficiency is found to be approximately 166%, 198%, and 192%, respectively. Similarly, YVO_4_: Tm^3+^ can be pumped by 473 nm. Upon the absorption of a visible photon around 473 nm, three NIR photons emitting at 1180, 1479, and 1800 nm can be obtained efficiently by the sequential three-step radiative transitions of Tm^3+^ [51]. Therefore, YVO_4_: Tm^3+^ can be studied as an efficient three-photon NIR quantum-cutting (QC) polycrystalline phosphor. As shown in Figure 7b, YVO_4_: Tm^3+^ presents NIR emissions peaked at 1180 nm and 1545 nm, which can be ascribed to the electronic transitions of ^3^F_3_→^3^H_6_ and ^1^G_4_→^3^F_3_. In addition, the ^3^H_4_→^3^F_4_ transition at 1479 nm would take place feasibly as the second-step emission of three-photon NIR-QC. Finally, in the third-step emission of three-photon NIR-QC, the populated ^3^F_4_ state would be released to the ^3^H_6_ ground state by emitting NIR photons peaking at 1800 nm. The researchers also mixed Bi^3+^ into the YNbO_4_: Tm^3+^ system as a sensitizer. Under ultraviolet (302 nm) excitation, the near-infrared luminescence of Tm^3+^ at 1820 nm (^3^F_4_→^3^H_6_) was increased by 151 times. The results are expected to conform to the current global condensed matter physics theme and contribute to the exploration of a new generation of environmentally friendly germanium solar cells. Moreover, Tm^3+^-activated phosphors can also be sensitized by Yb^3+^ to produce infrared to visible upconversion phosphors [52].

#### 4.1.6. NIR Phosphor Activated by Eu^2+^

As one of the most important activators, Eu^2+^ ion has been widely utilized to develop luminescence materials. Eu^2+^, with f–d parity, allows transitions and is constructive in reaching high internal quantum efficiencies and high absorption efficiencies. Generally, the emission band can be adjusted from the blue to the red region since the 5d–4f transition is highly associated with the crystal structure. BaMgAl_10_O_17_:Eu^2+^ [53] (BAM) has been explored as a blue phosphor with a wideband excitation band and a broadband emission peaked at 450 nm. As for the Eu^2+^-activated silicate phosphors, most of them own broad excitation bands ranging from 300 to 500 nm and present wide spectrum coverage. As early as 1968, Blasse et al. [54] reported the luminescence properties of M_2_SiO_4_:Eu^2+^ (M = Ca, Sr, Ba) phosphors. The silicate matrix corresponding to different alkaline earth metal cations would produce broadband emissions ranging from 505 nm to 575 nm. Thereafter, the continuous regulation of luminescence performance can be realized by adjusting the proportion of alkali earth metals and the concentration of Eu^2+^. Further, N^3-^ exhibits stronger covalence compared with O^2-^; the nitrogen-enriched crystal-field environment will produce an enhanced nephelauxetic effect, resulting in more significant effects on the 4f–5d transition of the activator ion. Consequently, the nitride red powders of M_2_Si_5_N_8_: Eu^2+^ and MAlSiN_3_: Eu^2+^ (M = Ca, Sr, Ba) have been reported by OSRAM, the National Institute for Material and Materials Research (NIMS), and Mitsubishi Chemical [55]. 

Recently, the development of Eu^2+^ ion-activated NIR luminescence materials has become an alternative to achieve high electro-optical conversion efficiency NIR LED light source devices. The NIR luminescence of Eu^2+^-doped phosphor has been reported in K_3_LuSi_2_O_7_: Eu^2+^ for the first time [56]. K_3_LuSi_2_O_7_: Eu^2+^ shows the emission centered at 740 nm with a full-width at half-maximum of 160 nm when it is pumped upon 460 nm blue light (Figure 8a). As shown in Figure 8b, K_3_LuSi_2_O_7_ offers one Lu site with 6-fold coordination and two different K sites for the possible occupation by Eu^2+^ ions. Herein, the unexpected NIR emission can be attributed to the selective site occupation of divalent europium in [LuO6] and [K2O6] polyhedrons with small coordination numbers and small average bond length, leading to the larger CFS in 5d energy levels and a larger redshift. According to the above work, it is found that when the Eu^2+^ ion occupies the crystallographic site with a smaller coordination number and shorter bond length, the energy level of the excited state Eu^2+^ will be severely split because of the enhanced crystal-field environment, which is conducive to realize the broadband NIR luminescence under blue light excitation. The work provides an alternative way to design novel NIR phosphors.

Subsequently, the same research group successfully developed near-infrared luminescence materials CaO: Eu^2+^ [57]. CaO, with a simple composition, is selected as the matrix material in this study. Herein, Ca^2+^ is 6-coordinated and the bond length of Ca-O is short, which can provide an enhanced crystal-field environment for the Eu^2+^ ion. In the early stage of development, CaCO_3_ and Eu_2_O_3_ are used as raw materials, and Eu^3+^ is directly reduced to Eu^2+^ by H_2_ at high temperatures. Broadband NIR luminescence, with a peak at 740 nm, can be obtained under the excitation of 450 nm blue light. However, the phosphor suffers from low luminescence performance due to the unsatisfactory reduction effect of Eu^3+^. Therefore, the novel carbon paper wrapping technology and oxygen defect repair strategy have been explored to reduce more Eu^3+^ in the lattice to Eu^2+^, consequently resulting in an enhanced luminescence performance of CaO: Eu^2+^, with the external quantum efficiency increasing from 17.9% to 54.7%. Compared with reported high-efficiency NIR phosphors, CaO: Eu^2+^ shows obvious advantages in both absorption efficiency and external quantum efficiency. Meanwhile, the thermal stability (@125 °C) of the CaO lattice was increased from 57% to 90% by utilizing the decomposition property of GeO_2_ under a high-temperature reduction atmosphere to repair the oxygen defect (Figure 9). Under 100 mA and 300 mA current drive, the photoelectric efficiency of NIR LED devices is up to 319.5 mW @ 23.4% and 766.1 mW @ 17.1%, respectively. Long-distance near-infrared-assisted photography shows its prospective application in the field as a high-power night-vision supplementary light source. The near-infrared phosphor prepared in this work provides an alternative for broadband high-efficiency NIR light sources.

### 4.2. NIR Phosphor Activated by Main-Group Metal

#### 4.2.1. NIR Phosphor Activated by Bismuth

Bismuth (Bi), as a major metal element, is located between nonmetallic elements and transition metal elements in the periodic table. Trivalent bismuth (Bi^3+^) has a characteristic absorption phenomenon in the n-UV region, which can be ascribed to a 6s→6p transition; therefore, it can effectively prevent the phenomenon of reabsorption [58]. Since its outermost electron has an exposed 6s shell, the luminescence performance of Bi^3+^ is easily affected by the nephelauxetic effect and crystal-field environment, and its luminescence behaviors can be adjusted from the ultraviolet region to the near-infrared region [58]. In addition, multiple valences of Bi elements can be used as luminescence centers, such as Bi^+^, Bi^2+^, Bi^5+^, and even Bi clusters composed of different valences. Bismuth-doped barium silicate glass has been reported to show broadband IR luminescence covering 1100 nm to 1600 nm wavelength region under a laser diode excitation at 808 nm (Figure 10a) [59]. The emission peak of the NIR luminescence exhibits at 1325 nm, with a full-width half-maximum (FWHM) of ~200 nm. Further, the luminescence intensity increases with Al_2_O_3_ content but decreases with increasing BaO content, suggesting that the NIR luminescence should be attributed to the bismuth Bi^2+^ or Bi^+^, and Al^3+^ ions play an indirect dispersing role for the luminescent centers. Bi_5_(GaCl_4_)_3_ is miscellaneous with the NaAlCl_4_ phase. Herein, the closo-deltahedral Bi_5_^3+^ cluster can absorb ultraviolet, visible, and infrared lights, and presents super-broad near-to-mid-infrared luminescence, ranging from 1 to 3 µm (Figure 10b) [60]. The above work will be a solid foundation for the development of practical devices that operate in the spectrum required for NMIR.

#### 4.2.2. NIR Phosphor Activated by Sn^2+^

Main-group elements with ns^2^ electronic structure, which can be excited by ultraviolet light, and produce the emission spectrum from ultraviolet to visible region. However, photoluminescence in the NIR is a very unusual feature among oxides containing main-group ions. 

Tin (Sn) is a typical main-group element, which can emit visible light at 300–700 nm. The emission of Sn^2+^ is affected by the local coordination environment since the valence state of the ns^2^ ion and the energy level of the p orbital are sensitive to the crystal-field environment. The Stokes shift tends to increase and the emission spectrum shifts toward the longer wavelength when the activator has an asymmetric coordination environment. At the same time, larger Stokes shifts can be obtained when luminescence involves relaxation from the so-called D excited-state energy level. Although the current study of the D excited-state energy level is not perfect, it is related to the charge transfer from the luminescence center to the surrounding matrix lattice to some extent. 

The luminescence properties of BaSnO_3_ have been investigated, and it is found that the material could obtain NIR luminescence centered at 905 nm at room temperature, and the luminescence properties were not related to doping activators, but to Sn^2+^ [61]. In order to explore the luminescence properties of Sn^2+^, a series of Ba_1−x_Sr_x_Sn_x_O_3_ have been synthesized. According to Figure 11, it is found that with the continuous increase in Sr composition, the octahedron centered on the Sn site is tilted, resulting in a decrease in the symmetry of the structure, and the position of the emission spectrum of Sn^2+^ moved toward the long-wave direction.

### 4.3. NIR Phosphor Activated by Transition Metals

Thanks to the transitions of outer shell d orbital electrons, the luminescence of transition metal (TM) ions-doped phosphors is affected by the local chemical environment and site symmetry, which consequently tunes the emission to the NIR range.

#### 4.3.1. NIR Phosphor Activated by Fe^3+^

Fe^3+^, an eco-friendly and chemically stable ion, has been known as a promising metal ion to obtain luminescence and studied for years, which is of intraconfigurational d–d transitions [62]. Fe^3+^ owns a d^5^ electron configuration and is an isoelectronic ion of Mn^2+^. However, due to the larger valence state of Fe^3+^, the crystal-field strength is stronger and the transition energy of ^4^T_1_(^4^G)→^6^A_1_(^6^S) is reduced; as a result, the emission wavelength of Fe^3+^-doped phosphor is longer than that of Mn^2+^-activated phosphors. In addition, the d–d transition of Fe^3+^ depends on its local environment, causing a tunable luminescence of Fe^3+^-doped phosphors. The previously reported emission wavelengths of Fe^3+^-doped phosphors are commonly located in the red and far-red light regions, which can be used for special fluorescent lamp applications. Recently, it has been investigated that when Fe^3+^ is doped into diverse hosts, it could exhibit NIR photoluminescence with an emission band in the 700–1000 nm range owing to the ^4^T_1_–^6^A_1_ transition. Presently, several Fe^3+^-activated phosphors have been developed for broadband NIR-emitting phosphor applications.

An unprecedented long-wavelength NIR luminescence of Fe^3+^ has been achieved in perovskite-derived host compound Sr_2−y_Ca_y_(InSb)_1−z_Sn_2z_O_6_ (Figure 12a). Under the excitation of 340 nm, Sr_2_InSbO_6_: Fe^3+^ exhibits a wide NIR luminescence band centered at 885 nm. With the replacement of Sr^2+^ with Ca^2+^, tuning the luminescence from 885 to 935 nm can be achieved, resulting in a significant increase in PL intensity (Figure 12b). In addition, the co-substitution of [In^3+^-Sb^5+^] by [Sn^4+^-Sn^4+^] further tunes the NIR luminescence from 935 nm to 1005 nm (Figure 12c). The tunable shift toward a longer wavelength can be ascribed to the strengthening of the crystal-field strength caused by lattice shrinkage (Figure 12d). Furthermore, the introduction of ions broadens the FWHM of NIR luminescence from 108 nm to 146 nm. The obtained Ca_2_InSbO_6_:Fe^3+^ phosphor reaches an ultra-high internal quantum efficiency (IQE) of 87% and external quantum efficiency (EQE) of 68%, respectively. A maximum NIR output power of 0.83 mW at 200 mA is achieved in the pc-LED fabricated by Ca_2_InSbO_6_:Fe^3+^, demonstrating the potential of Fe^3+^ activators in obtaining efficient NIR luminescence [63].

Moreover, NaScSi_2_O_6_: Fe^3+^ phosphor has been successfully achieved to obtain a NIR luminescence band centered at 900 nm under 300 nm ultraviolet light excitation, with a full-width at half-maximum (FWHM) of 135 nm and an internal quantum efficiency (IQE) of 13.3% (Figure 12e) [64]. As seen in Figure 12f, NaScSi_2_O_6_ is composed of SiO_4_ tetrahedra, ScO_6,_ and NaO_8_ polyhedra with corner-sharing. The Fe^3+^ activator is supposed to enter Sc^3+^ sites. The octahedral-coordinated Fe^3+^ ion is constructive in obtaining NIR luminescence (Figure 10g).

#### 4.3.2. NIR Phosphor Activated by Ni^2+^

The luminescence properties of the Ni^2+^-doped materials have been studied since 1963 when the fluorescence and the optical maser oscillation in MgF_2_: Ni^2+^ were observed. Ni^2+^ has a 3d^8^ electronic configuration, whose energy level is affected by the crystal field. The new energy-level configuration of d^8^ in octahedral sites is described by Tanabe–Sugano (TS) diagrams. In the case of a weak crystal field, ^3^T_2_ is the first excited-state energy level of Ni^2+^, while in the case of a strong crystal field, the first excited-state energy level is ^1^E. Because of its strong broadband emission in the NIR range, Ni^2+^ is often used as the luminescence center and is widely used in broadband optical amplifiers and tunable infrared lasers [65]. However, the emission of Ni^2+^-doped phosphors is hard according to blue light, making it hard to be used with blue chips.

It is found that Ni^2+^ ions have the ability to substitute for Ga^3+^ or Al^3+^ ions in the octahedral sites to achieve NIR luminescence. In the development of Ni^2+^-doped photoluminescent materials, La_3_Ga_5_GeO_14_: Ni^2+^ has been found to show NIR luminescence centered at 1430 nm, which can be assigned to the ^3^T_2_(^3^F)→^3^A_2_(^3^F) transition of the Ni^2+^ ion. Ni^2+^-doped LiGa_5_O_8_ has also been obtained [66], and dopant Ni^2+^ occupies a site of near-octahedral symmetry in LiGa_5_O_8_. The luminescence transition centered at 1220 nm, ranging from 1120–1800 nm. In addition, Yb^3+^ can be used as the sensitizer to improve the emission of Ni^2+^ by eight times [67] (Figure 13a). Zn_3_Ga_2_Ge_2_O_10_:Ni^2+^ has been developed to exhibit a broad short-wave infrared luminescence band, ranging from 1050 to 1600 nm and peaking at 1290 nm, which can be assigned to the ^3^T_2_→^3^A_2_ transition of Ni^2+^ locating in an octahedral environment. Meanwhile, the phosphor has two intense absorption bands centered at 370 and 620 nm, respectively (Figure 13b), This work leads the Ni^2+^ -activated NIR phosphors to find important applications in night-vision surveillance and biomedical imaging [68].

#### 4.3.3. NIR Phosphor Activated by Manganese

As a transition metal, Mn has a variety of valence states, such as Mn^2+^, Mn^3+^, Mn^4+^, Mn^5+^, and Mn^6+^. The various valence states of Mn can be used as the NIR luminescence center. However, the application is hampered by the variation in valence states, making it hard to obtain a stable valence state.

Mn^5+^ is a transition metal ion with a 3d^2^ electron configuration. The Mn^5+^ ion can occupy the tetrahedral coordination environment stably, where the ^1^E energy level is located under the ^3^T_2_ energy level, resulting in NIR luminescence in the region of 1000–1400 nm [69]. In addition, Mn^5+^-activated phosphors, such as phosphate and vanadate, show strong absorption and effective luminescence in the NIR region with a long fluorescence lifetime, which can be used in solid-state lasers and biological probes. However, until now, there have been no reports on the application of Mn^5+^-activated phosphors in fluorescence imaging, not only because pentavalent manganese is relatively rare in Mn but also because Mn^5+^-activated luminescence materials are difficult to synthesize through traditional chemical pathways. Mn^5+^ ions are unstable and often undergo valency reduction during the synthesis of an aqueous solution, resulting in the formation of MnO_2_ precipitation. Ba_3_(MO_4_)_2_: Mn^5+^ (M = V, P) nanoparticles with stable valence and controlled size have been successfully synthesized using a convenient two-step method for the first time by Qiu J. R. et al., which not only diminishes the reaction time but also reduces the reaction temperature [69] (Figure 14a). Since the ^1^E energy level locates below the ^3^A_2_ energy level when Mn^5+^ is located in the tetrahedral coordination, this consequently results in spin-forbidden emissions caused by a weak crystal-field strength. Therefore, both of the Ba_3_(MO_4_)_2_: Mn^5+^ (M = V, P) present a sharp emission band at 1190 nm originating from ^1^E→^3^A_2_ transition of Mn^5+^ ions under 808 nm excitation (Figure 14b). Additionally, two vibronic emission bands centered at 1250 nm and 1300 nm can be attributed to the vibronic transitions. According to the above work, Mn^5+^-doped luminescence materials exhibit remarkable NIR-II emissions around 1190 nm with a broad excitation band, suggesting the potential application of Mn^5+^ doping, as well as stimulating new ideas to prepare special valence ion-doped luminescence materials.

Mn^6+^ is a transition metal ion with a 3d^1^ electron configuration, which is stable in tetrahedral coordination environments. Generally, Mn^6+^-doped luminescence materials present a wide luminescence band in the NIR region, which can be considered a tunable and short-pulse laser [70]. However, Mn^6+^-doped luminescence materials are difficult to fabricate, requiring high reaction temperatures and long reaction times. In addition, it is difficult to control the particle size and morphology during the synthesis process, and the valence state of Mn tends to decrease, which will limit the application of these materials. BaSO_4_:Mn^6+^ luminescence material, with uniform size and morphology, has been successfully prepared for the first time by Qiu J. R. et al. through a fast liquid–solid solution route at room temperature [71] (Figure 14c). BaSO_4_: Mn^6+^ owns two broad bands ranging from 460 nm to 650 nm and 650 nm to 820 nm, originating from the O→Mn ligand-to-metal charge transfer (LMCT) transition and ^2^E→^2^T_2_ transition, respectively (Figure 14d). As shown in the PL spectrum, under 532 nm or 808 nm excitation, the phosphor exhibits a broad band centered at 1070 nm, extending from 900 to 1400 nm, which can be attributed to the d–d (^2^T_2_→^2^E) transition of Mn^6+^ ions in the MnO_4_^2−^ tetrahedra. This strategy opens a new gate for fabricating special-valence-state Mn ion-doped luminescence materials.

#### 4.3.4. NIR Phosphor Activated by Cr^3+^

As a transition metal ion, the Cr^3+^ ion has a special structure in that the outermost electron layer is not filled with electrons. The luminescence properties of Cr^3+^ can be strongly controlled by a crystal-field environment because of its special [Ar]3d^3^ electron configuration. The luminescence caused by Cr^3+^ is generally located in the region from deep red to NIR. In addition, Cr^3+^-doped luminescence material has attracted the attention of spectroscopists since the 1930s, owing to the linear emission of Cr^3+^ in the 680–720 nm spectral region in various luminescence materials. In 1958, spectroscopists explained the luminescence properties of ruby (Al_2_O_3_: Cr^3+^) for the first time through crystal-field theory [72]. Later in 1960, rubies were used in solid-state lasers [73]. According to the PL spectrum of rubies at 77K (Figure 15a), the Al_2_O_3_: Cr^3+^ has two linear luminescence at 692.9 nm and 694.3 nm, which are, respectively, called R1 and R2 lines, corresponding to the ^2^E_g_(t_2_^3^)→^4^A_2_(t_2_^3^) energy-level transition. With further research, the scientists found that when the Cr^3+^ ion is located in a compound with a weak crystal-field environment, such as gallium garnet, the ^4^T_2_ energy level takes the place of the ^2^E energy level for energy transition, consequently resulting in broadband NIR luminescence.

The Tanabe–Sugano diagram of Cr^3+^ is shown in Figure 15b, which can represent the relationship between the energy of the transition metal element level (y-coordinate E/B) and the crystal-field intensity (x-coordinate Dq/B) [74]. For the free ion Cr^3+^, its ground state is ^4^F. When Cr^3+^ is located in the octahedral coordination environment, the ^4^F energy level can be split into ^4^A_2_, ^4^T_1_, and ^4^T_2_ energy levels, among which ^4^A_2_ is the ground state. As the excited state, the ^4^P energy level can be split into ^4^T_1_, and the ^2^F energy level can be split into ^2^A_2_; meanwhile, the ^2^G energy level can be split into ^2^T_1_, ^2^T_2_, and ^2^E energy levels. There is an intersection between the ^4^T_2_ excited-state level and the ^2^E excited-state level, and the horizontal value of the intersection Dq/B is about 2.3. The left and right sides of the intersection point are divided into weak crystal fields and strong crystal fields.

The luminescence properties of Cr^3+^ ions are very sensitive to the surrounding crystal environment, the position of the ^4^T_2_ energy level of the Cr^3+^ ion depends on the crystal-field environment and coordination environment. (1) When the crystal-field strength is strong, the ^4^T_2_ energy level locates above the ^2^E energy level, and the luminescence of the Cr^3+^-doped phosphor originates from the ^2^E→^4^A_2_ transition. In addition, according to the Tanabe–Sugano diagram of Cr^3+^, The ^2^E energy level, and ^4^A_2_ energy level are almost parallel to each other, and do not change much with the variation in the crystal-field environment, consequently resulting in the sharp peak. In fact, the ^2^E energy level can be split into a double-energy level with a very small gap. Observed from the emission spectrum, it can be found that the two red chromatogram lines with relatively close wavelength positions are called the R1 and R2 lines. It is worth noting that the transition of the Cr^3+^ ion breaks the selection rule (∆S = 0) under the spin–orbital interaction and crystal field, and occurs between the dual and quadruple states. (2) When the Cr^3+^ states in a weak crystal field, the ^4^T_2_ energy level is located below the ^2^E energy level, consequently resulting in the broadband emission originating from the spin-allowed ^4^T_2_–^4^A_2_ transition. Herein, the peak position of broadband Cr^3+^ luminescence can be adjusted according to the strength of the crystal field. Scientists have also found a series of broadband luminescence of Cr^3+^-activated phosphors (Em = 650–1600 nm). (3) The ^4^T_2_ energy level locates near the ^2^E energy level when Cr^3+^ is located in the intermediate crystal field, the spin-forbidden ^2^E–^4^A_2_ (sharp) and the spin-allowed ^4^T_2_–^4^A_2_ (broad) transition occur simultaneously.

In conclusion, when Cr^3+^ ions serve as the luminescence center of phosphors, the tunable luminescence properties can be obtained by adjusting the composition of the lattice host, changing the local microstructure of phosphors, and finally changing the crystal-field environment. In addition, according to the excitation spectra of multiple phosphors activated by Cr^3+^ ions, it can be observed that the Cr^3+^-activated phosphors exhibit two wide excitation bands in the blue and red regions, attributed to the ^4^A_2_→^4^T_1_ and ^4^A_2_→^4^T_2_ transitions, respectively, suggesting its broad prospect in the field of NIR pc-LEDs excited by visible light chip. Recently, the research on NIR phosphors has mainly focused on Cr^3+^-activated NIR phosphors. Scientists have developed various Cr^3+^-doped NIR phosphors, including galliate/gallium-germanate, aluminate, silicate, borate, et. al.

Generally, the luminescence center is properly occupied in a crystallographic site with a similar radius, leading to less distortion of the lattice environment, consequently resulting in a more stable luminescence performance of phosphors. Considering the radius, valence state, and coordination environment, Cr^3+^ is more inclined to stably occupy the position of Ga^3+^ or Sc^3+^ with octahedral coordination, and the luminescence performance of the luminescent material is more stable. In the gallium/gallium-germanate, the ionic radius of Cr^3+^ is similar to that of Ga^3+^, especially the Ga^3+^ at the octahedral position. The doping of Cr^3+^ in the Ga^3+^ crystallographic site results in tiny lattice distortions, leading to an effective luminescence of Cr^3+^-doped gallium/gallium-germanates. When Cr^3+^ enters into the Ga^3+^ crystallographic site of LiGa_5_O_8_: Cr^3+^ with a stronger crystal-field strength, the phosphor exhibits a sharp line at 716 nm under ultraviolet light excitation, with a long afterglow property of more than 1000 h (Figure 15c) [75]. A similar luminescence property can also be observed in Zn_3_Ga_2_Ge_2_O_10_: Cr^3+^. The spectra of the sample show a narrow-band emission peak at 713 nm (Figure 15d), which can be attributed to the ^2^E–^4^A_2_ transition. In addition, the Ge plays the role of substituting Ga, which is conducive to the formation of traps. Therefore, the phosphor exhibits properties of afterglow. On the other hand, Cr^3+^ enters into the Ga^3+^ crystallographic site of SrGa_12_O_19_: Cr^3+^ with a weaker crystal-field strength, resulting in broadband NIR luminescence centered at 750 nm with 425 nm excitation (Figure 15d) [76]. The incorporation of Ga within SrAl_12_O_19_ magnetoplumbite-type structures induces broadband NIR luminescence in the range 650–1050 nm with the typical R line and additional broadband centered around 740–820 nm, which can be attributed to the spin-forbidden transitions of Cr^3+^–Cr^3+^ pairs. With increasing Ga^3+^ content, SrAl_12−x_Ga_x_O_19_: Cr^3+^ phosphors exhibit a blue-shift of the broadband NIR luminescence and enhanced R line [77]. Consequently, SrGa_12_O_19_: Cr^3+^ presents a high internal quantum efficiency of 95%. Furthermore, the phosphor has zero thermal quenching, which can be ascribed to increased absorption probability for the ^4^A_2_→^4^T_1_ and the depopulation of electrons from electron traps associated with defects. The luminescence performance demonstrates its potential for pc-NIR LED applications. Significantly, this work offers a novel pathway to design NIR phosphors with a Cr^3+^–Cr^3+^ pair.

Additionally, the luminescence spectrum can be broadened further when the luminescence spectra of luminescence centers with different crystallographic sites are superposed to each other. Three types of Cr^3+^ centers associated with the ^4^T_2_→^4^A_2_ transition can be found in Ca_3_Ga_2_Ge_3_O_12_. Herein, the configurations of three types of Cr^3+^ centers are octahedral, dodecahedral, and tetrahedral, peaking at about 749, 803, and 907 nm, respectively. Accordingly, the configurations of Ca^2+^, Ga^3+^, and Ge^4+^, where luminescence center Cr^3+^ occupies, are dodecahedral, octahedral, and tetrahedral, respectively. The order of the crystal-field strength is tetrahedral > octahedral > dodecahedral. Furthermore, the crystal-field environment of three types of Cr^3+^ centers, and the energy transfer will influence the luminescence of this phosphor. Consequently, the NIR luminescence is located at 650–1100 nm [78] (Figure 15e). Thanks to the energy transfer between different energy sites, the NIR luminescence can be tuned with the various luminescence center Cr^3+^ doping concentrations and excitation wavelengths.

According to La_3_Ga_5_GeO_14_: Cr^3+^, the phosphor shows super-broadband NIR luminescence centered at 980 nm under 460 nm excitation with a full-width at half-maximum of 330 nm (Figure 15f) [79]. Considering the radius and valence optimal principle, Cr^3+^ is determined to occupy the octahedral Ga1 and tetrahedral Ga3 crystallographic sites according to the ESR spectral analysis, XRD refinement, and first-principle calculations. The multiple crystallographic sites of the luminescence center Cr^3+^ successfully explain the reason for the ultra-wide luminescence spectrum of La_3_Ga_5_GeO_14_: Cr^3+^. However, the super-broadband NIR La_3_Ga_5_GeO_14_: Cr^3+^ phosphor suffers from poor luminescence intensity in applications. The luminescence intensity of Cr^3+^ can be enhanced by three times via Pr^3+^ doping. Herein, there is an obvious spectral overlap comparing the emission peaks of La_3_Ga_5_GeO_14_:Pr^3+^ and the excitation band of La_3_Ga_5_GeO_14_:Cr^3+^. Meanwhile, the luminescence intensity and lifetime of Pr^3+^ exhibit a decreasing trend with Cr^3+^ doping in La_3_Ga_5_GeO_14_: Pr^3+^, Cr^3+^, demonstrating the existence of the energy transfer from Pr^3+^ to Cr^3+^.

Scandate can offer a proper crystallographic site for Cr^3+^ to obtain NIR luminescence. ScBO_3_: Cr^3+^ has been found to provide the [ScO6] lattice with an octahedral coordination environment for a Cr^3+^ luminescence center to obtain broadband NIR luminescence centered at 800 nm under 460 nm excitation with a quantum yield of 65% (Figure 16a) [80]. Whereafter, ScBO_3_: Cr^3+^ is coated with a blue chip to achieve a pc-LED with a NIR light output power of ~26 mW and an energy conversion efficiency of 7%, demonstrating that ScBO_3_: Cr^3+^ NIR phosphor can be effectively used in NIR LED devices in food detection and other fields. La(Sc, M)_3_B_4_O_12_: Cr^3+^ (M = Ga, Y, Ca-Si) has been developed for pc-LEDs for various applications. Herein, LaSc_3_B_4_O_12_, possessing a monoclinic structure with *C2/c* (No. 15) space group, is composed of scandium–oxygen octahedrons, boron–oxygen trigonal-planar units, and lanthanum–oxygen trigonal prisms (Figure 16b) [81]. Thereinto, [ScO_6_] octahedron provides an ideal coordination polyhedron for the Cr^3+^ activator to substitute. Consequently, the phosphor offers broadband NIR luminescence centered at 874 nm under blue light excitation, which can be ascribed to the ^4^T_2_–^4^A_2_ transition (Figure 16c). To adjust the luminescence performance for diversity applications, the scandium–oxygen octahedron has been designed to suffer variation. Ga^3+^, with a smaller radius, is designed to substitute Sc^3+^ to obtain targeted phosphor LaSc_2.93−*y*_Ga*_y_*B_4_O_12_ Cr^3+^ (0 ≤ *y* ≤ 1.5) by chemical composition modification. The substitution of Sc^3+^ by Ga^3+^ in LSGB Cr^3+^ leads to the enhanced crystal field and decreasing polyhedron distortion of the structure, consequently resulting in the tunable luminescence behavior. On one hand, broadband emissions corresponding to the ^4^T_2_–^4^A_2_ transition shifts from 871 to 824 nm, accompanied by an enhanced sharp emission peak at 689 nm attributed to ^2^E–^4^A_2_ with Ga^3+^ doping, which can be ascribed to an enhanced crystal field caused by lattice shrinkage (Figure 16d). On the other hand, the phosphors exhibit better thermal quenching behavior due to decreasing polyhedron distortion with increasing Ga^3+^ content. As a result, the LSGB: 0.07Cr^3+^ (*y* = 0.6) is demonstrated to be a promising candidate for blue-pumped security monitoring LEDs. Inversely, the substitution of Sc^3+^ by larger Y^3+^ in LaSc_2.93−*x*_Y*_x_*B_4_O_12_(LSYB): 0.07Cr^3+^ (0 ≤ *x* ≤ 1.3) induces a lattice expansion, leading to a weakened crystal-field environment in the LSYB (0 ≤ *x* ≤ 1.3) lattice host, consequently resulting in a spectral shift from 871 nm to 883 nm owing to the ^4^T_2_–^4^A_2_ transition (Figure 16e) [82]. In addition, the structure evolution caused by Y^3+^ doping modifies the band gap of LSYB (0 ≤ *x* ≤ 1.3), which will then improve the quantum efficiency and thermal quenching behavior of LSYB: 0.07Cr^3+^. Moreover, considering that Si is usually considered a stable element to ensure high luminescence intensity, a nonequivalent co-substitution of Sc^3+^-Sc^3+^ by Ca^2+^-Si^4+^ is used to extend the luminescence performance of LaSc_3−2*x*_Ca*_x_*Si*_x_*B_4_O_12_(LSCSB): Cr^3+^ (0 ≤ *x* ≤ 1.0) for enriching its versatility [83]. Herein, the octahedral position, where Cr^3+^ is accommodated, shows a lattice expansion due to the Ca^2+^-Si^4+^ doping, which provides a weakened crystal-field environment. Consequently, the phosphors present a regular shift of luminescence spectra from 871 to 880 nm (Figure 16f).

Moreover, a consequence of phosphors, such as oxides and fluorides has been developed. The garnet-type material has a rigid crystal structure, which can effectively inhibit the nonradiative relaxation of the luminescence center. Consequently, it is expected to obtain NIR luminescence with high efficiency and good thermal stability. Accordingly, various garnet-type NIR phosphors have been designed. The garnet of type Lu_3_Sc_2_Ga_3_O_12_: Cr^3+^ has been studied as a model, while chemical unit co-substitution is used to develop a novel NIR phosphor. As known, the Cr^3+^ ion owns a d–d forbidden-transition, resulting in low absorption efficiency. In order to solve the above problem, the chemical co-substitution of [Sc^3+^-Ga^3+^] by [Mg^2+^-Si^4+^] has been designed to regulate the distortion degree of the local structure of the Cr^3+^ ion. The ionic radius of Mg^2+^ and Si^4+^ is smaller than that of Sc^3+^ and Ga^3+^, the lattice presents a shrinkage with the chemical co-substitution, and the ionic property of the Cr–O bond can be enhanced. With increasing [Mg^2+^-Si^4+^] content, the spectra present a redshift from 706 nm for *x* = 0 to 765 nm for *x* = 0.6, as well as a broadened half-peak width (FWHM) to 176 nm. So far, a series of broadband NIR phosphors with luminescence behavior has been obtained. Further, the chemical co-substitution of [Sc^3+^-Ga^3+^] by [Mg^2+^-Si^4+^] reduces the local symmetry of the [CrO_6_] octahedron, resulting in a partially unbanned ^4^T_2_–^4^A_2_ transition, which enhances the absorption efficiency and external quantum efficiency of phosphor. The optimized phosphor owns high thermal stability, maintaining the luminescence intensity at room temperature to 150 °C compared with that at 25 °C. With a similar structure, the garnet of type Ca_2_LuZr_2_Al_3_O_12_: Cr^3+^ is found to exhibit broadband NIR luminescence ranging from 750 to 820 nm [84]. However, the phosphor suffers from low luminescence intensity. Considering the PLE spectra of Ca_2_LuZr_2_Al_3_O_12_: Cr^3+^, Ce^3+^ can be used as a sensitizer to improve the luminescence intensity of Cr^3+^. It is found that the emission spectra of Ca_2_LuZr_2_Al_3_O_12_: Ce^3+^ overlapped with the excitation spectra of Ca_2_LuZr_2_Al_3_O_12_: Cr^3+^, and the luminescence intensity and lifetime of Ce^3+^ exhibit a decreasing trend with Cr^3+^ doping in Ca_2_LuZr_2_Al_3_O_12_: Ce ^3+^, Cr^3+^, demonstrating the existence of the energy transfer from Ce ^3+^ to Cr^3+^. Finally, the luminescence intensity of Cr^3+^ can be improved by the energy transfer from Ce^3+^.

Recently, the luminescence center of Cr^3+^ was located in the intermediate crystal-field environment in the fluorides K_3_AlF_6_ and K_3_GaF_6_, exhibiting a broadband centered at 750 nm [85]. As a fluoride, the phosphor exhibits lower phonon energy than oxides. Therefore, the d–d transition of Cr^3+^ is less affected by electron–phonon coupling, which is beneficial for the good thermal stability of fluoride phosphor. Compared to K_3_GaF_6_: Cr^3+^, K_3_AlF_6_: Cr^3+^ has a larger distortion in the luminescence center, consequently resulting in a strengthened zero phonon line due to the ^4^T_2_→^4^A_2_ transition. The phosphor presents good thermal stability, maintaining 87.3% of the initial luminescence intensity at 150 °C. Under a driving current of 100 mA, the NIR light-emitting diodes reach a photoelectric efficiency of 9.315%, demonstrating their potential in night-vision imaging systems. Further, a much simpler fluoride, ScF_3_:Cr^3+^ [86], has been developed to obtain NIR luminescence. The phosphor presents a broadband emission ranging from 700 to 1100 nm, peaking at 853 nm, with the full-width at half-maximum (FWHM) of 140 nm. As for thermal stability, the phosphor exhibits a good thermal quenching behavior, which maintains 85.5% luminescence intensity at 150 °C compared with that at 25 °C. The luminescence behavior indicates its potential toward blue-based chip LEDs.

Herein, a series of Cr^3+^-doped NIR phosphors has been summarized to understand the research (Table 1).

According to the above works, various promising phosphors for the application of blue-pumped NIR pc-LED toward diversified applications have been designed. Moreover, a consequence of regular patterns has been summarized to provide an efficient pathway to design and optimize the Cr^3+^-doped NIR phosphors with proper luminescence properties. Generally, the barycenter shift (ε_c_), owing to the nephelauxetic effect, as well as the crystal-field variation due to the distorted polyhedron, can verify the spectral shift of phosphor. According to Cr^3+^-doped phosphors, the strength of the covalent bond is in direct proportion to the nephelauxetic effect, leading to an increasing barycenter shift (ε_c_) (Figure 17a), while the shrinkage of the crystallographic site where the luminescence center occupies will enhance the crystal-field strength, resulting in a blue-shift (Figure 17b). As for the luminescence intensity and thermal quenching behavior, the doping of the proper sensitizer will transfer the energy to the activator, resulting in enhanced luminescence intensity (Figure 17c). Decreasing the polyhedron distortion of the host lattice results in a higher-rigidity of structure, finally causing larger activation energy (Ea) and better thermal stability (Figure 17d).

#### 4.3.5. NIR Phosphor Activated by Cr^4+^

As a transition metal element, Cr has a variety of valence states, among which the ions that emit light in the near-infrared region are Cr^3+^ and Cr^4+^. Similar to Cr^3+^, the Cr^4+^ ion has a special structure in that the outermost electron layer is not filled with electrons. The luminescence properties of Cr^4+^ can be strongly controlled by a crystal-field environment because of its special [Ar]3d^2^ electron configuration. The Tanabe–Sugano diagram of Cr^4+^ in a tetrahedral coordination environment is shown in Figure 18a, which can represent the relationship between the energy of the transition metal element level (y-coordinate energy/B) and the crystal-field intensity (x-coordinate 10Dq/B). For the free ion Cr^4+^, its ground state is ^3^F. When Cr^4+^ is located in the tetrahedral coordination environment, the ^3^F energy level can be split into ^3^A_2_, ^3^T_1_, and ^3^T_2_ energy levels, among which ^3^A_2_ is the ground state. As the excited state, the ^3^P energy level can be split into ^3^T_1_; meanwhile, the ^1^D energy level can be split into ^1^E. There is an intersection between the ^3^T_2_ excited-state level and the ^1^E excited-state level. The left and right sides of the intersection point are divided into a weak crystal field and a strong crystal field [100,101].

When the Cr^4+^ ion is located in a weak crystal-field environment, the ^3^T_2_ excited-state level is located under the ^1^E energy level, and the Cr^4+^-doped phosphor emits light originating from the ^3^T_2_→^3^A_2_ transition. Thanks to the large lattice relaxation of this transition, it exhibits wideband luminescence under the action of the self-selected permissible transition. Further, it is found that the E/B value of the ^3^T_2_ excited-state level is changed with variation in the crystal-field environment. Therefore, the luminescence behavior of Cr^4+^-doped phosphors can be adjusted in a crystal-field environment. When the Cr^4+^ ion is located in a strengthened crystal-field environment, the ^3^T_2_ excited-state level is located above the ^1^E energy level, and the Cr^4+^-doped phosphor emits light originating from the ^1^E→^3^A_2_ transition. In addition, the ^1^E energy level is almost parallel to the horizontal axis, that is, the ^1^E energy level and ^3^A_2_ energy level are almost parallel to each other, and do not change much with the variation in the crystal-field environment, consequently resulting in the peak emission of Cr^4+^-doped phosphors. Cr^4+^-doped luminescence materials have been studied in crystalline materials. Ca_2_GeO_4_:Cr^4+^ crystalline material has been reported to obtain NIR emissions with full-width at half-maximum (FWHM) of 201 nm, peaking at 1317 nm [102].

Recently, Cr^4+^-doped phosphors have been generally used to obtain NIR luminescence with longer emission wavelengths, ranging from 1100 nm to 1300 nm. Ca_2_Al_2_SiO_7_: Cr^4+^ aluminosilicate nanoparticle has been developed as a NIR-to-NIR nanothermometer [103]. As shown in Figure 18b,c, the luminescence material shows an absorption band ranging from 600 nm to 800 nm, which can be assigned to the ^3^A_2_→^3^T_2_ transition, while the absorption bands peaking at 430 nm and 1050 nm can be attributed to the ^3^A_2_→^3^T_1_ and ^3^A_2_→^3^T_2_ transition, respectively. The absorption band ranging in the 600–800 nm region is located in luminescence material (BTW-I: 650–950 nm), potentially offering the possibility of being used as a NIR-to-NIR nanothermometer. The luminescence material emits the NIR broadband emission ranging from 1100 nm to 1600 nm, centered at 1230 nm, under the excitation of 730 nm, while the emission band is just covering the BTW-II (1050–1350 nm). In addition, as the Cr^4+^ is located in a relatively weak crystal-field environment, the Ca_2_Al_2_SiO_7_: Cr^4+^ presents a tunable luminescence behavior from 1230 nm to 1238 nm with a temperature variation. The luminescence behavior demonstrates its applicability in the application of temperature sensing.

A multifunctional and dual-excited NIR phosphor Mg_14_Ge_5_O_24_: Cr^3+^, Cr^4+^ has been developed to be used toward night vision, agriculture, and bio-applications [104]. Herein, Mg_14_Ge_5_O_24_ owns a carious of Ge^4+^ sites, [GeO_6_] is coordinated with six O^2-^ ions to form octahedrons, and the others are coordinated with four O^2-^ ions to form [GeO_4_] tetrahedrons. The co-existence of [GeO_6_] octahedrons and [GeO_4_] tetrahedrons in the Mg_14_Ge_5_O_24_ provides favorable and suitable coordination for Cr^3+^ and Cr^4+^. Considering the charge balance and ionic radius, Cr^3+^ can be located in a 6-coordinated Ge^4+^ site, while it is possible for Cr^4+^ to occupy a 4-coordinated Ge^4+^ site. The content of Cr^3+^ and Cr^4+^ can be changed with increasing Cr-doping. When the concentration of Cr-doping is low, Cr^3+^ is doped in a weak crystal-field environment, exhibiting a broadband NIR emission in the region of 650–1100 nm, with full-width at half-maximum (FWHM) of 266 nm, originating from the spin-allowed ^4^T_2_→^4^A_2_ transition of Cr^3+^. With increasing Cr-doping, more and more Cr^4+^ is preferentially located in four-coordinated Ge^4+^ sites. Consequently, the phosphor presents broadband NIR emissions from 1100 nm to 1600 nm with an FWHM of 256 nm, which can be assigned to the ^3^T_2_→^3^A_2_ transition of Cr^4+^. The emission of Cr^3+^ under blued-light excitation is a coincidence within the NIR I region; meanwhile, the emission of Cr^4+^ under NIR light excitation is a coincidence within the NIR II region. The phenomenon of NIR I region excitation and NIR II region emission appears, demonstrating its great application potential in the field of NIR broadband LED for biological detection (Figure 18d).

## 5. Conclusions and Outlook

In a word, the NIR light source shows great application prospects in modern agriculture, security monitoring, food safety, and other fields. At present, NIR pc-LEDs, using mature visible light chips combined with phosphor, are a promising NIR light source, which has become an international research and development hotspot. As one of the key materials of pc-LEDs, the phosphor can directly determine the luminescence efficiency and spectral continuity of NIR LED devices. A consequence of the works has been focused on the design of high-performance phosphors and their applications. All of the above achievements provide theoretical and experimental guidance for the design and development of NIR phosphors. Nevertheless, studies on NIR phosphor have just started with very limited available materials. Several challenges remain in this field, including but not limited to the following:

(1) Exploring effective methods for designing novel NIR phosphors. At present, the trial-and-error method is generally used in luminescence material designing; there is a lack of novel design methods and high-performance materials. In addition, the phosphors with emission peaks ranging from 700 nm to 900 nm are rich in variety. However, broadband NIR phosphors with luminescence peaks longer than 900 nm are particularly scarce, and are in urgent need of development to meet applications in diverse fields.

(2) Further enhancement of the comprehensive performance of NIR phosphors in applications. In general, the external quantum efficiency of the visible phosphor used in LED is over 65%. By contrast, NIR phosphors suffer from low external quantum efficiency of <35% owing to a larger Stokes shift compared to that of visible phosphors under blue/UV excitation, which will hinder their prospects in applications. In addition, NIR pc-LEDs are generally driven by high power in practical applications, resulting in a stricter requirement on the thermal stability of NIR phosphors. Hence, it is an urgent requirement to improve the comprehensive performance of NIR phosphors.

(3) Developing the application technology in different fields. Because of the low luminescence efficiency of NIR phosphor compared with visible light, the NIR pc-LEDs own the special situation of a high ratio of phosphor to glue, resulting in a low light extraction rate and low light transmittance rate. Various institutions have focused on the related phosphors and light-emitting devices [105,106,107,108]. In terms of application, OSRAM has preliminarily explored the NIR light source for the detection of food ingredients [109,110,111]. GRIREM Advanced Materials Co., Ltd. has created a simulated-sunlight light source ranging from visible to NIR wavelength, which can initially meet the application requirements. Nevertheless, the world is still in the initial stage of application technology development, which is not mature. Therefore, it is an urgent requirement to develop packaging and application technology for different application scenarios.

At present, in the field of traditional w-LED, the industrial scale of our country takes the top spot in the world; however, the core intellectual property rights are always controlled by others. In the future, we hope to occupy the pioneer advantage of material design, preparation technology, and application technology in the field of nonvisible light, and become the leading state in this field.

## Figures and Tables

**Figure 1 materials-16-03145-f001:**
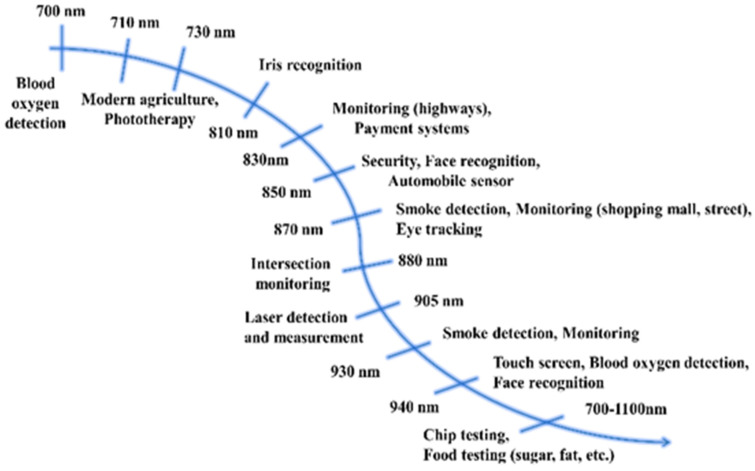
Applied analysis of far-red/NIR light.

**Figure 2 materials-16-03145-f002:**
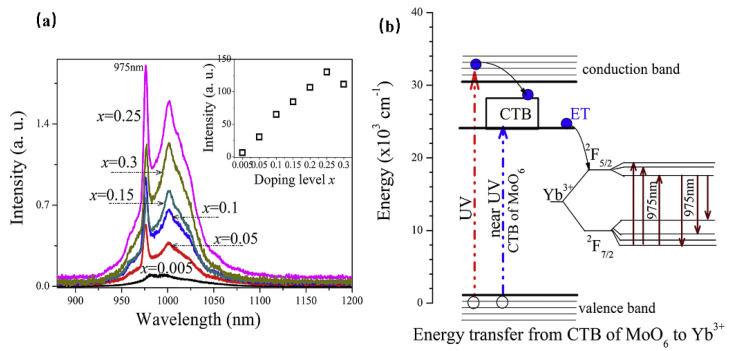
(**a**) The PL spectra of Y_4−x_Yb_x_CdMo_3_O_16_ (x = 0–0.3) phosphors (Ex = 400 nm and Em = 980 nm). (**b**) The scheme of the suggested energy transfer from the Y_4_CdMo_3_O_16_ host to the Yb^3+^ ions.

**Figure 3 materials-16-03145-f003:**
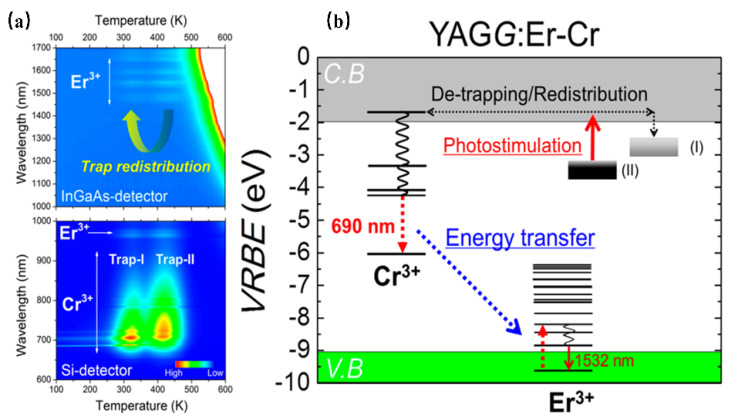
(**a**) Contour mapping of the thermoluminescence intensity after UV illumination as a function of emission wavelength (λ) and temperature (T) of the YAGG: Er–Cr ceramic samples. (**b**) Energy-level diagram for Cr^3+^ and Er^3+^ in a garnet host.

**Figure 4 materials-16-03145-f004:**
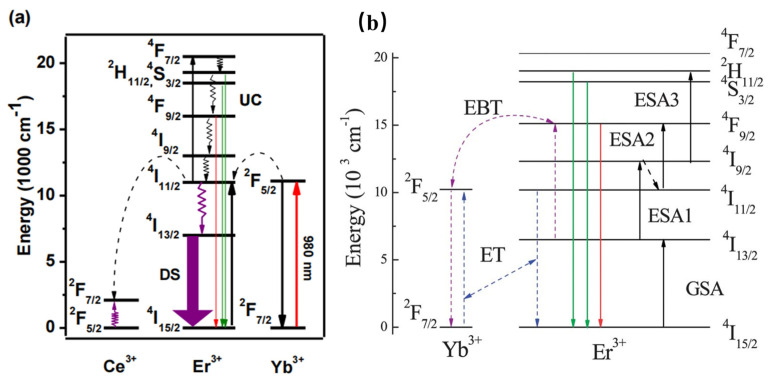
(**a**) Schematic mechanism for the energy-transfer process in the UC emission of Er^3+^ ions. (**b**) Energy-level diagram for Er^3+^ and Yb^3+^ ions.

**Figure 5 materials-16-03145-f005:**
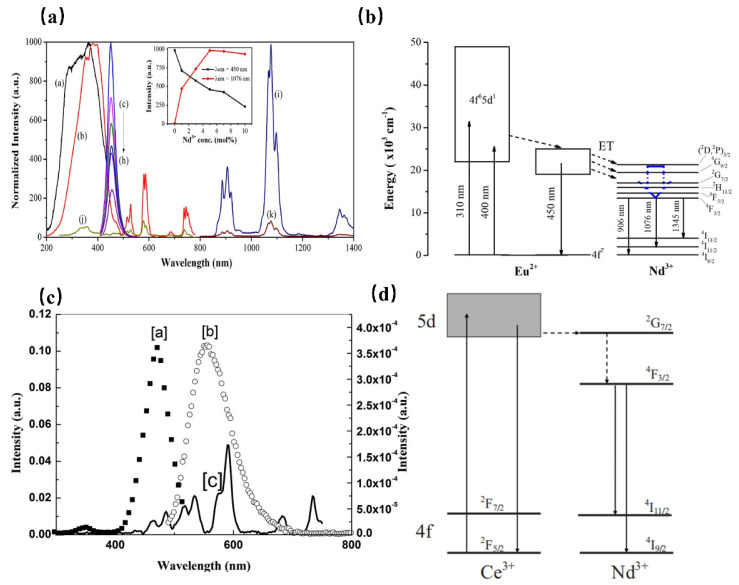
(**a**) PLE (a, Em = 450 nm and b, Em = 1076 nm) and PL (c–i, Ex = 400 nm) spectra of the Ca_2_PO_4_Cl: Eu^2+^, Nd^3+^ samples. PLE (j, λem = 1076 nm) and PL (k, λex = 400 nm) spectra of the Ca_2_PO_4_Cl: Nd^3+^ (5 mol%) sample. (**b**) Schematic energy-level diagram for Eu^2+^ and Nd^3+^ in the Ca_2_PO_4_Cl phosphor. (**c**) Fluorescence spectra of YAG: Ce and YAG: Nd. (a, PLE spectrum of YAG: Ce Em = 550 nm, b, PL spectrum of YAG: Ce Ex = 476 nm, and c, PLE spectrum of YAG: Nd Em = 1064 nm). (**d**) Schematic energy-level diagram for YAG: Ce^3+^, Nd^3+^.

**Figure 6 materials-16-03145-f006:**
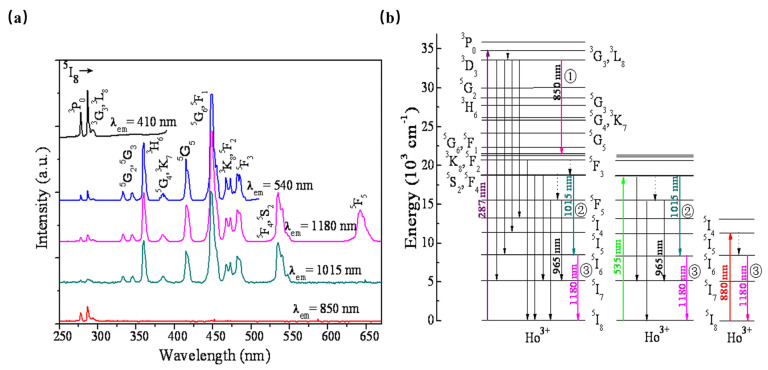
(**a**) PLE spectra monitored at various PL wavelengths of β-NaYF_4_:Ho^3+^ phosphor. (**b**) Schematic energy-level diagram for Ho^3+^ showing the concept of sequential three-step NIR-QS under excitation of 287, 535, and 808 nm, respectively.

**Figure 7 materials-16-03145-f007:**
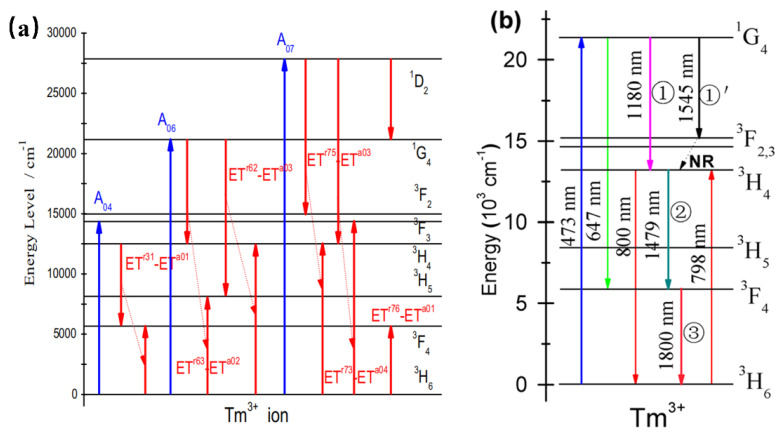
(**a**) Schematic diagram of the energy-level structure and quantum-cutting process of Tm: YNbO_4_ powder phosphor. (**b**) Energy-level diagram of Tm^3+^ schematically illustrating the ET mechanisms, mainly involving the sequential three-step three-photon NIR-QC.

**Figure 8 materials-16-03145-f008:**
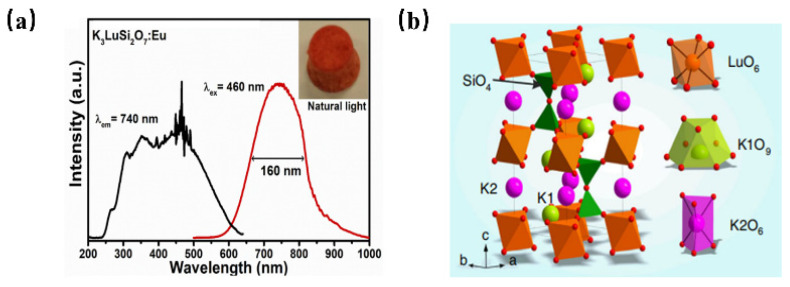
(**a**) PL and PLE spectra of K_3_LuSi_2_O_7_: Eu^2+^ (Ex = 460 nm, Em = 740 nm) and (**b**) crystal structure of K_3_LuSi_2_O_7_.

**Figure 9 materials-16-03145-f009:**
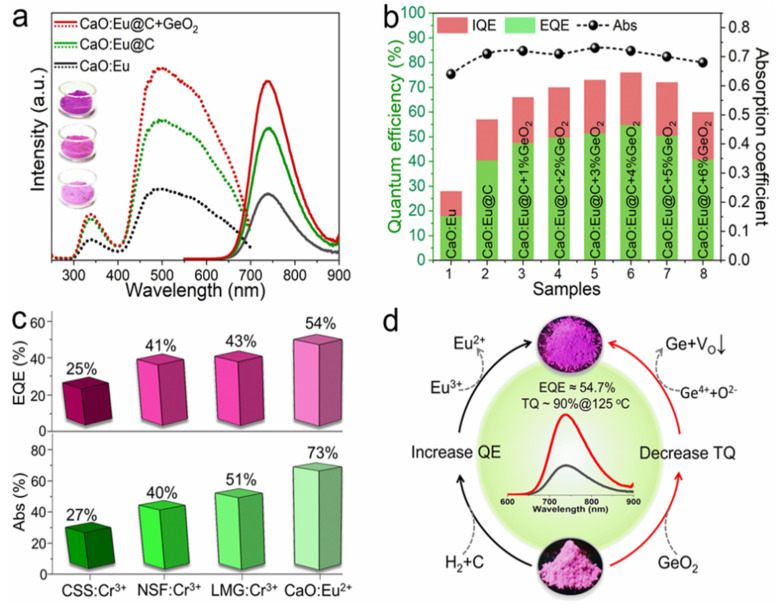
(**a**) Excitation and emission spectra of Eu^2+^ phosphor under different treatment processes (Ex = 450 nm and Em = 740 nm). (**b**) Diagram of internal/external quantum efficiency and absorption efficiency. (**c**) Comparison of properties of several NIR phosphors. (**d**) CaO: Schematic diagram of enhancement mechanism of luminescence and thermal stability of Eu^2+^ phosphor.

**Figure 10 materials-16-03145-f010:**
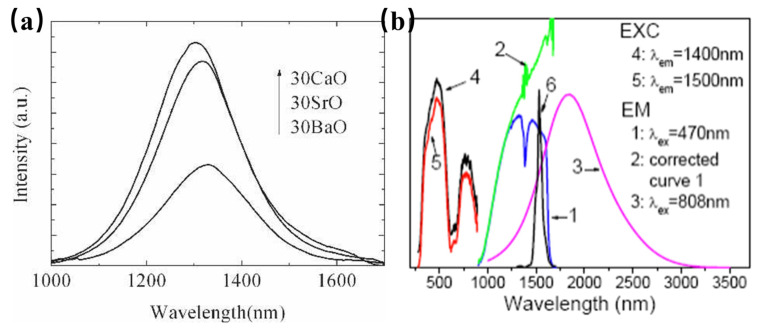
(**a**) Fluorescence spectra of glasses 65SiO_2_·30RO·5Al_2_O_3_. 1Bi_2_O_3_ (R = Ca, Sr, and Ba), excited by 808 nm laser diode (Em = 1325 nm). (**b**) Emission and excitation spectra of Bi_5_(GaCl_4_)_3_ at room temperature.

**Figure 11 materials-16-03145-f011:**
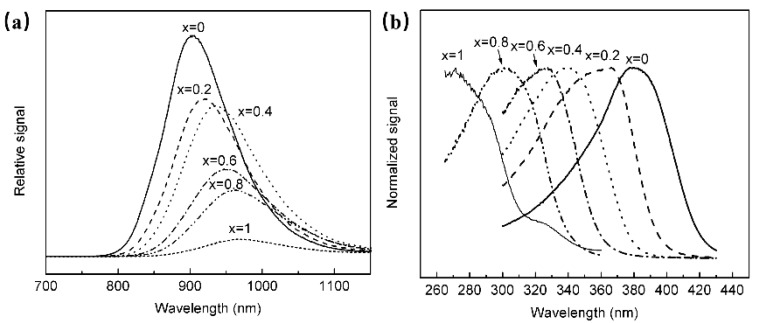
(**a**) PL spectra of Ba_1−x_Sr_x_Sn_x_O_3_ (Ex = 380 nm). (**b**) PLE spectra of Ba_1−x_Sr_x_Sn_x_O_3_ (Em = 905 nm).

**Figure 12 materials-16-03145-f012:**
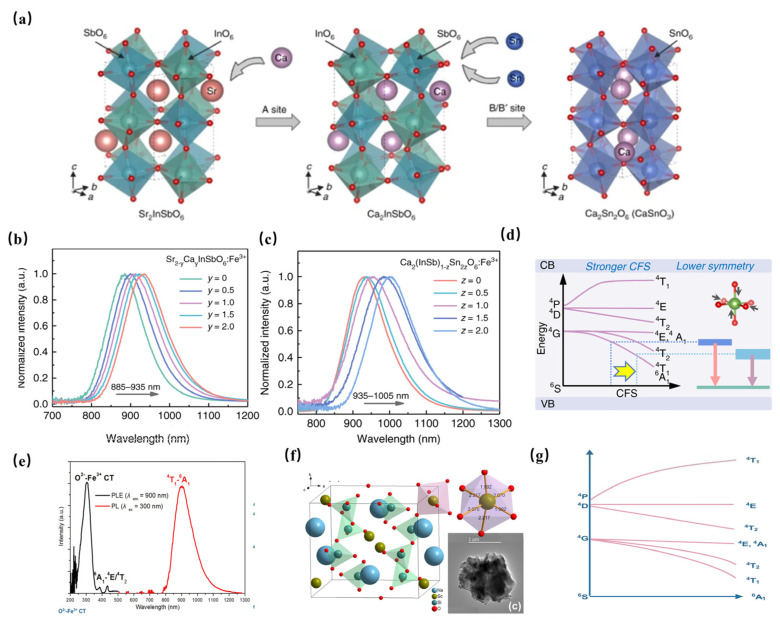
(**a**) Structure characterizations of Sr_2−*y*_Ca*_y_*(InSb)_1−*z*_Sn_2*z*_O_6_:Fe^3+^ (Ex = 340 nm). (**b**) Normalized PL spectra of Sr_2−*y*_Ca*_y_*InSbO_6_:Fe^3+^(*y* = 0–2) (Ex = 340 nm). (**c**) Normalized PL spectra of Ca_2_(InSb)_1−*z*_Sn_2*z*_O_6_:Fe^3+^ (*z* = 0–1). (**d**) Schematic diagram of the overall PL tuning mechanism [63]. (**e**) PL and PLE spectra of NaScSi_2_O_6_: Fe^3+^ (Ex = 300 nm, Em = 900 nm) [64]. (**f**) The structural diagram of NaScSi_2_O_6_ [64]. (**g**) The schematic configurational diagram of octahedral-coordinated Fe^3+^ ions [64].

**Figure 13 materials-16-03145-f013:**
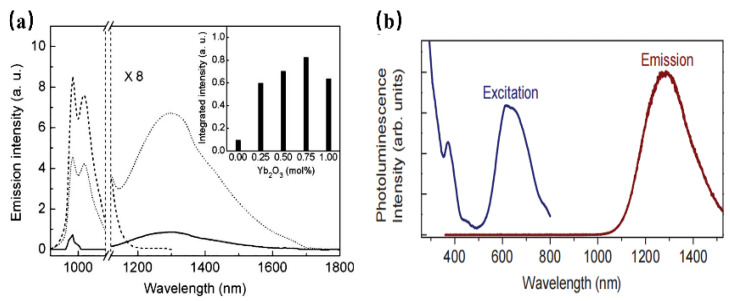
(**a**) Emission spectra of Ni^2+^ (solid curve), Yb^3+^ (dashed curve), and Yb^3+^ /Ni^2+^ (dotted curve)-doped LiGa_5_O_8_ under 980 nm excitation. The inset shows the dependence of the integrated intensity of Ni^2+^ emissions on the Yb_2_O_3_ concentration. (**b**) PLE and PL spectra of Zn_3_Ga_2_Ge_2_O_10_:Ni^2+^ (Ex = 370 nm and Em = 1290 nm).

**Figure 14 materials-16-03145-f014:**
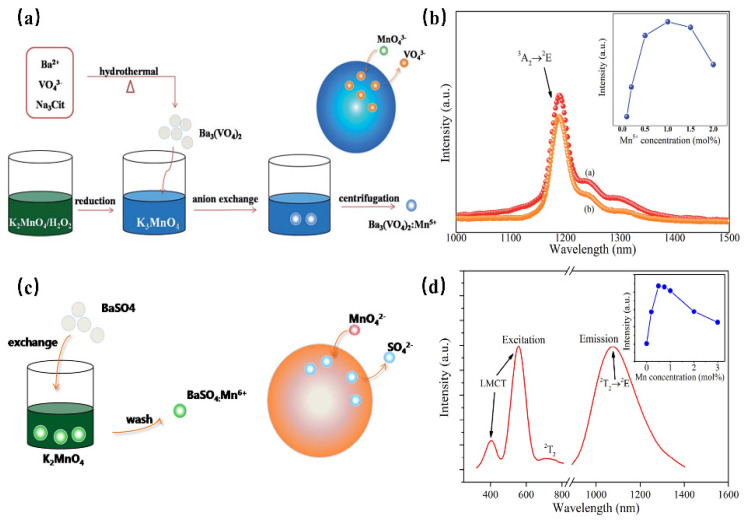
(**a**) Schematic illustration of anion exchange procedure for synthesizing Mn^5+^-doped Ba_3_(VO_4_)_2_ [69]. (**b**) NIR emission spectra of Ba_3_(MO_4_)_2_: Mn^5+^ (M = V, P) phosphors under 808 nm excitation. Inset: concentration dependence of NIR emission intensity of Ba_3_(VO_4_)_2_: Mn^5+^ phosphor (Em = 1190 nm). (**c**) Schematic illustration of anion exchange procedure for the synthesis of BaSO_4_:Mn^6+^. (**d**) PLE and PL spectra of BaSO_4_:Mn^6+^(Ex = 532 nm and Em = 1070 nm).

**Figure 15 materials-16-03145-f015:**
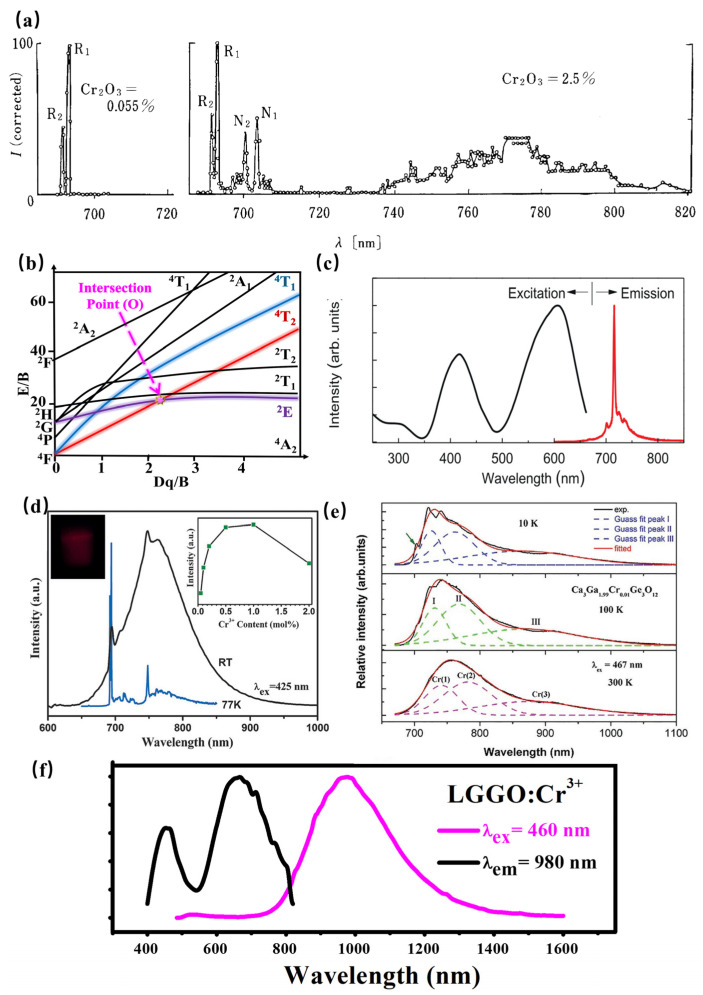
(**a**) PL spectrum of rubies at 77K. (**b**) Tanabe–Sugano diagram of Cr^3+^. (**c**) PL spectra of LiGa_5_O_8_: Cr^3+^ (Ex = 400 nm and Em = 716 nm). (**d**) PL spectra of SrGa_12_O_19_: Cr^3+^ (Ex = 425 nm and Em = 713 nm). (**e**) PL spectra of Ca_3_Ga_2_Ge_3_O_12_: Cr^3+^ (Ex = 267 nm). (**f**) PL spectra of La_3_Ga_5_GeO_14_: Cr^3+^ (Ex = 460 nm and Em = 980 nm).

**Figure 16 materials-16-03145-f016:**
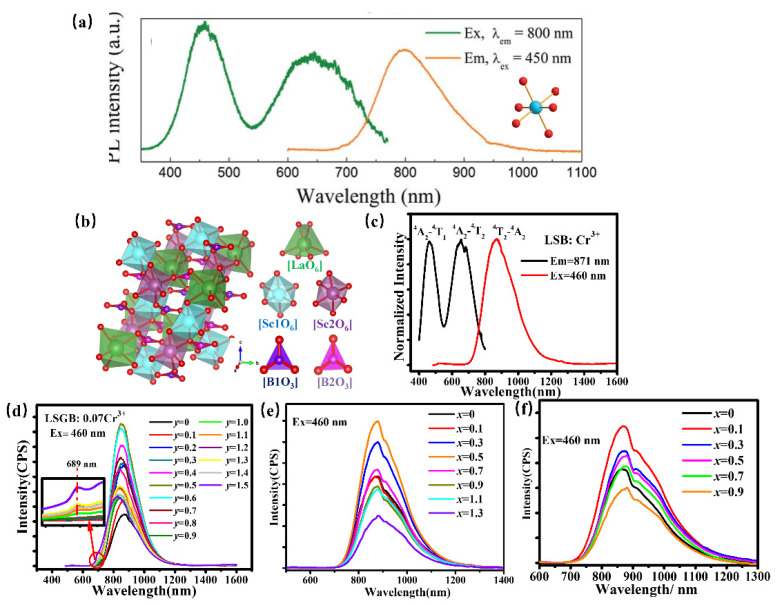
(**a**) PLE and PL spectra of ScBO_3_: Cr^3+^ and the coordination environment of [ScO6] octahedron (Ex = 450 nm and Em = 800 nm) [80]. (**b**) The crystal structure of LaSc_3_B_4_O_12_ (**c**) PLE and PL spectra of LaSc_3_B_4_O_12_: Cr^3+^ (Ex = 460 nm and Em = 871 nm). (**d**) PL spectra of LaSc_2.93−*y*_Ga*_y_*B_4_O_12_: 0.07Cr^3+^ (0 ≤ *y* ≤ 1.5) (The inset showing the variation at 689 nm) (Ex = 460 nm) (**e**) PL spectra of LaSc_2.93−*x*_Y*_x_*B_4_O_12_: 0.07Cr^3+^ (0 ≤ *x* ≤ 1.3) (Ex = 460 nm). (**f**) PL spectra of LaSc_3−2*x*_Ca*_x_*Si*_x_*B_4_O_12_(LSCSB): Cr^3+^ (0 ≤ *x* ≤ 1.0) (Ex = 460 nm).

**Figure 17 materials-16-03145-f017:**
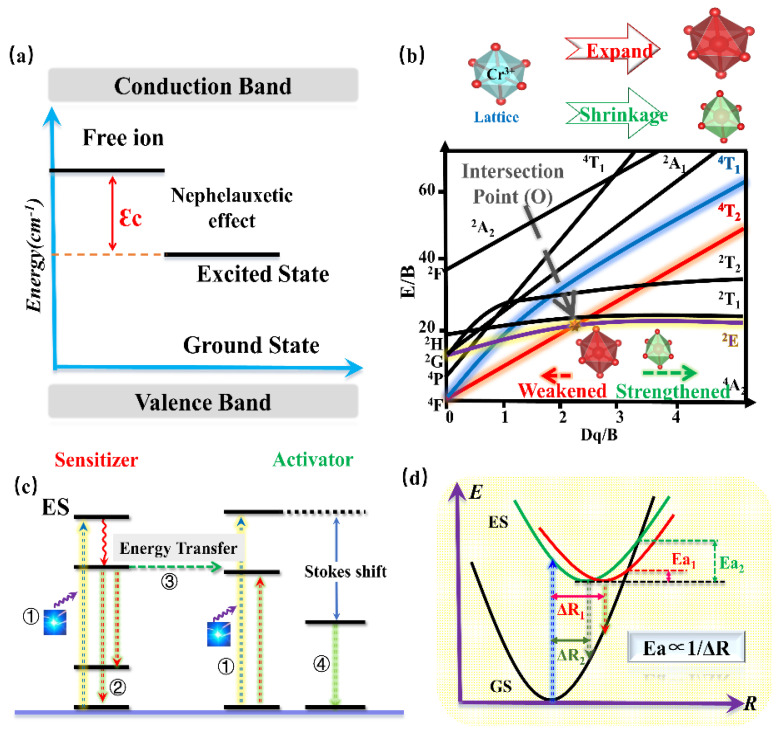
(**a**) Energy level diagram for Cr^3+^.ε_c_ is a barycenter shift. (**b**) Crystal field strength of Cr^3+^ according to different lattice sizes. (**c**) Energy transfer model. (**d**) The configurational coordinate models.

**Figure 18 materials-16-03145-f018:**
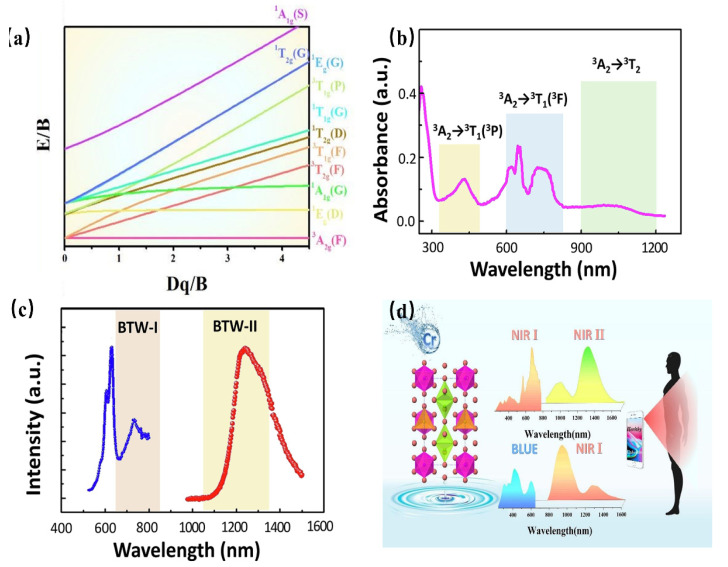
(**a**) Energy-level diagram for Cr^4+^. (**b**) Absorption spectra of Ca_2_Al_2_SiO_7_: Cr^4+^. (**c**) PL spectra of Ca_2_Al_2_SiO_7_: Cr^4^ (Ex = 710 nm and Em = 1230 nm). (**d**) PLE and PL spectra of Mg_14_Ge_5_O_24_: Cr^3+^, Cr^4+^ (Ex = 420 nm and Em = 1240 nm).

**Table 1 materials-16-03145-t001:** A series of Cr^3+^-doped NIR phosphors.

Phosphors	λ_max_ (nm)	FWHM (nm)	Efficiency	Photoelectric Efficiency/Light Output	Reference
ScF_3_:Cr^3+^	853	140	IQE = 45%		[76]
K_2_NaScF_6_: Cr^3+^	765	100	IQE = 74%		[87]
Na_3_AlF_6_: Cr^3+^	720	95	QY = 75%		
ScBO_3_: Cr^3+^	800	120	QY = 72.8%	39.11 mW @350 mA	[88]
LiScP_2_O_7_: Cr^3+^	880	170	IQY = 74%		[89]
Ca_2_LuHf_2_Al_3_O_12_: Cr^3+^	785	145			[90]
Ca_2_LuZr_2_Al_3_O_12_: Cr^3+^	752	117	IQE = 69.1% EQE = 31.6%	750–820 nm 4.1%; 350–1100 nm 8.51% @20 mA	[63]
LiInSi_2_O_6_: Cr^3+^	840	143	QY = 75%	17.8% @100 mA/ 51.6 mW @100 mA 3V	[91]
Mg_7_Ga_2_GeO_12_:Cr^3+^	800	266	IQE = 86% EQE = 37%		[92]
Li_2_Sr_2_Al(PO_4_)_3_	823	178	IQE = 61%		[93]
Ca_3_Sc_2_Si_3_O_12_: Cr^3+^	770	110	IQE = 92.3%	109.9 mW @520 mA	[94]
LaMgGa_11_O_19_: Cr^3+^	770	138	IQE = 82.6% EQE = 42.5%		[95]
Ca_2_LuScGa_2_Ge_2_O_12_: Cr^3+^	800	150		1.213 mW @0.4%	[96]
La_2_MgZrO_6_: Cr^3+^	825	210	IQE = ~56% EQE = ~17.9%		[97]
Lu_3_Sc_2_Ga_3_O_12_: Cr^3+^	722	73	QE = ~60%		[98]
YAl_3_(BO_3_)_4_: Cr^3+^	720	110	QY = ~86.7%	the light output power of ~50.6 mW and energy conversion efficiency of ~17.4% at 100 mA drive current	[99]

## Data Availability

No new data were created.

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
