# Peer review of "Research Progress and Development of Near-Infrared Phosphors"

_materials, 2023, doi:10.3390/ma16083145_

Round 1

Reviewer 1 Report

In 3. Introduction of light emitting device, the explanations of 3.3 SiC heating element is not suitable for this part.

Please make sure the introduction in Section 4 should contain all activators- doped phosphors. For example, in 4. Research status of NIR phosphor, the Tm3+-doped phosphors were not described; however, Ni2+ was presented. Furthermore, the Er3+-Yb3+ co-doped phosphors, which have been widely studied, should be added in this draft.

Author Response

Response to Reviewer 1 Comments

Point 1: In 3. Introduction of light emitting device, the explanations of 3.3 SiC heating element is not suitable for this part.

Response 1: Thank you for pointing this out.

The explanations of 3.3 SiC heating element has been deleted. The changes are highlighted with the yellow background.

Point 2: Please make sure the introduction in Section 4 should contain all activators- doped phosphors. For example, in 4. Research status of NIR phosphor, the Tm3+-doped phosphors were not described; however, Ni2+ was presented. Furthermore, the Er3+-Yb3+ co-doped phosphors, which have been widely studied, should be added in this draft.

Response 2: Thank you for pointing this out.

The description of Tm3+-doped phosphors and Er3+-Yb3+ co-doped phosphors have been added. The changes are highlighted with the yellow background.

Reviewer 2 Report

The comments are attached.

Author Response

Response to Reviewer 2 Comments

The review article “Research Progress and Development of Near-infrared Phosphors” reports the overview of various ions used for emission in NIR spectral range. Several comments on this review article are given below.

Point 1: The English should be improved since there are many grammar mistakes, for instance, missing space between number and unit (980nm instead of 980 nm).

Response 1: Thank you for your careful suggestion.

We have checked our paper, and all the modifications are marked in yellow.

Point 2: In many places the font size is larger than the rest of the text. This should also be corrected.

Response 2: Thank you for your careful advice.

We have checked our paper to modify the font size.

Point 3: Overall, the manuscript is well-written and easy to read and understand. I would suggest increasing the size of the figures to make the more readable.

Response 3: Thank you for your useful advice.

We have increased the size of the figures to make the more readable.

Point 4: It is strange that the introduction part does not have any references.

Response 4: Thank you for your careful discussion.

The Introduction part mainly introduces the application fields of NIR light with different emission wavelength. The above information mainly comes from market research conducted by our research group. We have added the related references in the paper, the changes have been marked in yellow.

Point 5: Title of Figure 2. Section (a) shows the emission spectra of Yb3+; however, the title says that the PL excitation spectra are given. Please check and correct.

Response 5: We are sorry for making such mistake.

We have corrected the title, and the changes have been marked in yellow.

Point 6: Section 4.1.3. Nd3+ in various hosts can also be sensitized with Ce3+ . The authors could mention this too. Besides, Eu2+ doped compounds can also efficiently sensitize Nd3+  as was reported by S. Möller et al. (https://doi.org/10.1016/j.jlumin.2015.11.040). Therefore, the Eu2+ does not necessarily need to emit in the blue to sensitize Nd3+ emission. I am sure there are also other references out there to support this.

Response 6: Thank you for your useful discussion.

We have discussed this part, and the changes have been marked in yellow.

Point 7:Title of Figure 7. It should be ‘a) Excitation and emission spectra’ instead of ‘a) Excitation emission spectra’

Response 7: We are sorry for making such mistake.

We have corrected the title, and the changes have been marked in yellow.

Point 8: Line 374. Tin should be used instead of stannum.

Response 8: We have modified “Tin” in the paper.

Point 9: Lines 608-612. Herein, the configurations of three types of Cr3+ centers are dodecahedral, octahedral, and tetrahedral, peaking at about 749, 803, and 907 nm, respectively. Accordingly, the configurations of Ca2+, Ga3+, and Ge4+, where luminescence center Cr3+ occupy, are dodecahedral, octahedral, and tetrahedral, respectively. Based on the order of the crystal field strength: tetrahedral > octahedral > dodecahedral, Cr3+ is sited…. Authors write that Cr3+ in tetrahedral sites emit at 907 nm. This is correct. However, the comparison in line 612 (tetrahedral > octahedral > dodecahedral) is not correct, since it suggests that tetrahedral coordination generates the strongest crystal field. Please check and correct.

Response 9: Thank you for your careful advice.

The disscusion for this part has been modified, and the changes have been marked in yellow.

Point 10: There are two figures name Figure 14. Please check and correct.

Response 10: We are sorry for making such mistake.

We have corrected the number of Figure.

Point 11: This review article cites 69 papers. From my point of view, it is not enough, especially for such a broad topic.

Response 11: Thank you for pointing this out.

We have added our paper with more studieds.

Point 12: I would also suggest for authors to include some tables where the general phosphor information is compared, like chemical composition, activator, excitation wavelength, emission wavelength, FWHM, efficiency, etc. This is very useful representation of the data for the reader.

Response 12: Thank you for pointing this out.

We have added Table 1 in thepaper. The changes have been marked in yellow.

Reviewer 3 Report

General Comment:

The manuscript from Gao Tongyu, Chen Guantong, Liu Yuanhong, Liu Ronghui, Zhuang Weidong and Ma Xiaole presents a review of the major advances and developments in near-infrared phosphors. An analysis of the fields of application of infrared light emitting devices and the main families of emissive compounds (with lanthanides, main group elements or/and transition metals) are presented. Being a review, this work is rather well written, understandable and well organized, the different parts of the review highlight the weak and strong points of each phosphor family. In my opinion, without being a specialist in IR phosphorescent compounds except those with lanthanides, i find that this review in adequacy with the journal “Materials”. Some improvement could be realized before that this manuscript is published. I recommend publication after minor revisions.

Comments and Minor points:

1- My main point of restraint comes from the figures, which are probably all for the most part figures from the literature manuscript, they are unfortunately very often too small and not readable. The authors should definitely try to remedy this by enlarging/modifying/simplifying the proposed figures. In figure 14, the correspondences of figures 14e and f do not appear in the title of the figure. There are two figures numbered 14, so the number of the second figure numbered 14 should be changed and the number of figure 15 should be changed to 16. The changes in the numbers of figures 14, 15 and 16 will also lead to a change in the text of the manuscript. Still from the point of view of figures, it is essential to systematically show the excitation and/or emission wavelengths in the legend and/or in the figures directly whenever a PL or PLE measurement is presented.

2- There are some typographical errors in the manuscript, it is necessary to reread the document in detail in order to eliminate the small errors (missed space, 1E instead of 1E,…). Concerning the numbering of the quotations, it would be preferable to use the classical numbers and not the Roman numbers in order to be in adequacy with the references at the end of the manuscript. A number of references (15, 16, 17, 19, 20, 21, 22, 23) have a strange format with [] at the beginning of the line just after the number. Crystallographic terms should be indicated in italics (Pn-3n, C2/c)

Also watch out for the different font and font size used, it could be that this impression comes from the formatting of the pdf file, but it seems that sometimes the text changes font and/or font size.

3- An important point is that the authors present work without citing their own research, which is an extremely positive point. However, out of 69 references, even if I don't doubt the important part of the Chinese community to contribute to this field, only about twenty references come from non-chinese groups, and I think that the authors must absolutely remedy this by citing more works coming from North and South America, Europe, and even Africa, and from Asia but in a broader sense.

Author Response

Response to Reviewer 3 Comments

General Comment:

The manuscript from Gao Tongyu, Chen Guantong, Liu Yuanhong, Liu Ronghui, Zhuang Weidong and Ma Xiaole presents a review of the major advances and developments in near-infrared phosphors. An analysis of the fields of application of infrared light emitting devices and the main families of emissive compounds (with lanthanides, main group elements or/and transition metals) are presented. Being a review, this work is rather well written, understandable and well organized, the different parts of the review highlight the weak and strong points of each phosphor family. In my opinion, without being a specialist in IR phosphorescent compounds except those with lanthanides, i find that this review in adequacy with the journal “Materials”. Some improvement could be realized before that this manuscript is published. I recommend publication after minor revisions.

Point 1: My main point of restraint comes from the figures, which are probably all for the most part figures from the literature manuscript, they are unfortunately very often too small and not readable. The authors should definitely try to remedy this by enlarging/modifying/simplifying the proposed figures. In figure 14, the correspondences of figures 14e and f do not appear in the title of the figure. There are two figures numbered 14, so the number of the second figure numbered 14 should be changed and the number of figure 15 should be changed to 16. The changes in the numbers of figures 14, 15 and 16 will also lead to a change in the text of the manuscript. Still from the point of view of figures, it is essential to systematically show the excitation and/or emission wavelengths in the legend and/or in the figures directly whenever a PL or PLE measurement is presented.

Response 1: Thank you for pointing this out.

We have increased the size of the figures and checked the number of Figures to make the more readable. What’s more, we have show the excitation and/or emission wavelengths in the caption of Figures. The changes have been marked in yellow.

Point 2: There are some typographical errors in the manuscript, it is necessary to reread the document in detail in order to eliminate the small errors (missed space, 1E instead of 1E,…). Concerning the numbering of the quotations, it would be preferable to use the classical numbers and not the Roman numbers in order to be in adequacy with the references at the end of the manuscript. A number of references (15, 16, 17, 19, 20, 21, 22, 23) have a strange format with [] at the beginning of the line just after the number. Crystallographic terms should be indicated in italics (Pn-3n, C2/c)

Response 2: Thank you for your careful advice. We are sorry for making such mistake.

We have checked our paper and corrected the errors.

Point 3: An important point is that the authors present work without citing their own research, which is an extremely positive point. However, out of 69 references, even if I don't doubt the important part of the Chinese community to contribute to this field, only about twenty references come from non-chinese groups, and I think that the authors must absolutely remedy this by citing more works coming from North and South America, Europe, and even Africa, and from Asia but in a broader sense.

Response 2: Thank you for your discussion.

We have cited a series of works from foreign countries.

Round 2

Reviewer 2 Report

I think the authors did a great job improving the manuscript, and I support its publication in its current state.

Author Response

Thank you!